# Evolutionary divergence in the conformational landscapes of tyrosine vs serine/threonine kinases

**Joan Gizzio[1,2†], Abhishek Thakur[1,2†], Allan Haldane[1,3], Ronald M Levy[1,2]\***

[1]Center for Biophysics and Computational Biology, Temple University, Philadelphia, United States; [2]Department of Chemistry, Temple University, Philadelphia, United States; [3]Department of Physics, Temple University, Philadelphia, United States

**\*For correspondence:**
ronlevy@temple.edu

[†]These authors contributed equally to this work

**Competing interest:** The authors declare that no competing interests exist.

**Abstract** Inactive conformations of protein kinase catalytic domains where the DFG motif has a "DFG-out" orientation and the activation loop is folded present a druggable binding pocket that is targeted by FDA-approved 'type-II inhibitors' in the treatment of cancers. Tyrosine kinases (TKs) typically show strong binding affinity with a wide spectrum of type-II inhibitors while serine/threonine kinases (STKs) usually bind more weakly which we suggest here is due to differences in the folded to extended conformational equilibrium of the activation loop between TKs vs. STKs. To investigate this, we use sequence covariation analysis with a Potts Hamiltonian statistical energy model to guide absolute binding free-energy molecular dynamics simulations of 74 protein-ligand complexes. Using the calculated binding free energies together with experimental values, we estimated free-energy costs for the large-scale (~17–20 Å) conformational change of the activation loop by an indirect approach, circumventing the very challenging problem of simulating the conformational change directly. We also used the Potts statistical potential to thread large sequence ensembles over active and inactive kinase states. The structure-based and sequence-based analyses are consistent; together they suggest TKs evolved to have free-energy penalties for the classical 'folded activation loop' DFG-out conformation relative to the active conformation, that is, on average, 4–6 kcal/mol smaller than the corresponding values for STKs. Potts statistical energy analysis suggests a molecular basis for this observation, wherein the activation loops of TKs are more weakly 'anchored' against the catalytic loop motif in the active conformation and form more stable substrate-mimicking interactions in the inactive conformation. These results provide insights into the molecular basis for the divergent functional properties of TKs and STKs, and have pharmacological implications for the target selectivity of type-II inhibitors.

## Editor's evaluation

This important paper provides a convincing mechanism for the relative binding specificity of Type II inhibitors to kinases. The combination of a sequence-derived Potts model with experimental dissociation constants and calculated free energies of binding to the DFG-out state is highly compelling and goes beyond the current state-of-the-art. Given the importance of kinases in pathophysiological processes, the results will be of interest to a broad audience and, in addition, the combination of computational methods can be applicable to a wide variety of other biophysical processes that involve conformational rearrangements.

## Introduction

The human genome contains approximately 500 eukaryotic protein kinases which coordinate signaling networks in cells by catalyzing the transfer of a phosphate group from ATP to serine, threonine, or tyrosine residues (*Manning, 1995*; *Modi and Dunbrack, 2019a*). The GO (gene ontology) database identifies 351 (~70%) of these enzymes as serine/threonine kinases (STKs) and 90 (~18%) as tyrosine kinases (TKs). STKs are an ancient class of protein kinases that predate the divergence of the three domains of life (bacteria, archaea, eukaryote) (*Stancik et al., 2018*), whereas TKs are a more recent evolutionary innovation, having diverged from STKs about 600 million years ago during early metazoan evolution (*Miller, 2012*; *Sebé-Pedrós et al., 2016*). Kinases are important therapeutic targets in a large number of human pathologies and cancers. Both TKs and STKs share a striking degree of structural similarity in their catalytic domains, owing to evolutionary selective pressures that preserve their catalytic function; in particular, the location and structure of the ATP binding site are highly conserved which raises significant challenges for the design of small-molecule ATP-competitive inhibitors that are both potent for their intended target(s) and have low off-target activity for unintended kinase targets. The latter is referred to as the 'selectivity' of an inhibitor, a property which is difficult to predict and control but is nonetheless very important for developing drugs with minimal harmful side effects.

A particular class of ATP-competitive kinase inhibitors which were proposed to have a high potential for selectivity are called 'type-II inhibitors' which only bind when the kinase adopts an inactive 'DFG-out' conformation. 'DFG' (Asp-Phe-Gly) refers to a conserved catalytic motif located at the N-terminus of an ~20 residue-long 'activation loop' that is highly flexible and controls the activation state of the kinase and the structure of the substrate binding surface. The precise arrangement of catalytic residues and the structural organization of large regulatory elements, such as the activation loop and nearby 'αC-helix', are strongly coupled to the conformation of the DFG motif and the DFG-1 residue preceding it, which is well described by regions on the Ramachandran map occupied by the Asp, Phe, and DFG-1 residues (beta-turn, right-handed alpha-helix, left-handed alpha-helix) and the $\chi_1$ rotamer state of the DFG-Phe sidechain (trans, gauche-minus, gauche-plus). Recently, Dunbrack and co-workers identified eight major conformational states in the Protein Data Bank (PDB) based on these metrics (*Modi and Dunbrack, 2019b*). The most common state, which is evolutionarily conserved in all kinases, corresponds to the active 'DFG-in' conformation. In this conformation all structural requirements for catalysis are typically met, e.g., a complete hydrophobic spine, a salt bridge between the conserved β3-Lys and αC-Glu residues, and an extended activation loop which forms the substrate binding surface. Inactive kinases in the PDB are most frequently seen in an 'Src-like inactive' conformation where the DFG is 'in', but the αC-helix is swung outward, breaking the β3-Lys → αC-Glu salt bridge and disassembling the hydrophobic spine. Disassembly of the hydrophobic spine caused by αC-helix rotation increases the cavity volume around the DFG-Phe residue, allowing it to pass through the Src-like inactive conformation and completely 'flip' from DFG-in to DFG-out (*Levinson et al., 2006*; *Shan et al., 2013*). The classical DFG-out conformation, targeted by type-II inhibitors, displays a highly reorganized activation loop that is folded away from the αC-helix, projecting toward solvent or forming stable secondary structure and substrate-mimicking interactions. We refer to these states of the activation loop collectively as 'folded', to describe its ~17 Å reorganization relative to the active 'extended' conformation, wherein the substrate binding surface has been 'folded up' toward the kinase N-terminal lobe and away from the αC-helix.

In both TKs and STKs, the activation loop undergoes this large-scale conformational change when the DFG motif flips from the active 'DFG-in' conformation to the classical DFG-out conformation. The DFG flip swaps the positions of DFG-Phe and DFG-Asp, opening a hydrophobic 'back pocket' that is connected to the conserved ATP binding site through the 'gatekeeper' residue. Type-II inhibitors typically have a long chemical fragment that allows them to bind across the gatekeeper and form interactions with residues in the back pocket. In contrast, type-I inhibitors (the majority of kinase drugs) occupy the ATP pocket but not the back pocket and can bind to either DFG-in or DFG-out. For these reasons, it has been proposed that type-II inhibition holds greater potential for the design of highly selective drugs (*Vijayan et al., 2015*; *Davis et al., 2011*; *Anastassiadis et al., 2011*); it has been shown that different kinase sequences have different propensities to adopt DFG-out in the absence of inhibitor (*Haldane et al., 2016*; *Hari et al., 2013*), and the DFG-out back pocket has been suggested to have a lesser degree of sequence and structural homology between kinases (*Liu and Gray, 2006*). However, the notion that type-II inhibitors developed to-date are more selective than type-I inhibitors

has been brought into question (*Zhao et al., 2014*; *Klaeger et al., 2017*), suggesting that further consideration of the energetic contributions described above is required.

In order to fully exploit the target-selective potential of type-II inhibitors it is necessary to understand the underlying sequence-dependent principles that control the conformational preferences of their kinase targets, and the extent to which this has been diversified by evolution. This can, in principle, be directly approached using free-energy simulations to estimate the reorganization free-energy required for different kinases to adopt DFG-out, although this is computationally very expensive and of uncertain reliability for conformational changes involving large-scale loop reorganizations, such as the ~17 Å 'folding' of the activation loop that accompanies the transition from active DFG-in to the inactive, classical DFG-out state. To accommodate this limitation, we employ modern sequence-based computational methods to characterize the conformational selection process over the entire kinome and combine the sequence-based results with structure-based free energy simulations with the goal of identifying evolutionarily divergent features of the energy landscape that control the preference of individual kinases for the active (DFG-in) vs inactive (DFG-out 'folded activation loop') states. To this end, we report evidence that TK catalytic domains have a molecular evolutionary bias that shifts their conformational equilibrium toward the inactive 'folded activation loop' DFG-out state in the absence of activation signals. In contrast, STKs as a class have a more stable active conformation which is favored over the DFG-out state due to sequence constraints in the absence of other signals.

As described below, our analysis of a previously published kinome-wide assay suggests that TKs have properties which privilege the binding of type-II inhibitors in comparison to STKs, which leads us to hypothesize an evolutionary divergence in their conformational energy landscapes. To investigate this, we used a Potts Hamiltonian statistical energy model derived from residue-residue covariation in a multiple sequence alignment (MSA) of protein kinase sequences to probe the active DFG-in ↔ classical DFG-out conformational equilibrium as previously described (*Haldane et al., 2016*). Using an approach that involves 'threading' a large number of kinase sequences onto ensembles of active DFG-in and classical DFG-out structures from the PDB and scoring them using the Potts Hamiltonian, we are able to view the evolutionary divergence in TK and STK conformational landscapes. This calculation only probes the free-energy difference between the active DFG-in and classical DFG-out conformations, and by construction does not consider alternative conformations (e.g. 'Src-like inactive') that might be important for analyzing the type-II binding pathway. As discussed below, the Potts calculations from this two-state model correlate well with the free-energy cost to adopt the classical DFG-out conformation.

To validate our results, we used the Potts statistical energy threading calculations to guide target selection for a set of more computationally intensive free-energy simulations. These simulations use type-II inhibitors as tools to probe kinase targets that have already reorganized to DFG-out, allowing us to estimate the free-energy of reorganization ($\Delta G_{reorg}$) as the excess between the absolute binding free-energy (ABFE) calculated from simulations and the standard binding free-energy measured experimentally *in vitro*, which already includes the cost to reorganize. Although our methods avoid sampling the conformational change directly, we show how important structural determinants of the conformational change can be identified by analyzing residue-pair contributions to the Potts threading calculations, enabling us to reason about the molecular evolutionary basis for the differences in conformational behavior observed for TKs and STKs.

## Results

### Insights into the sequence-dependent conformational free-energy landscape

The binding of type-II inhibitors is achieved once a protein kinase has reorganized to the DFG-out with activation loop folded conformation (classical DFG-out). We sought initial insight into the conformational equilibrium from type-II binding data available publicly in the form of literature-reported dissociation constants ($K_d$). From the binding assay reported by *Davis et al., 2011*, we report a 'hit' where an inhibitor binds to a kinase with $K_d \leq 10$ μM. Using this criterion, a type-II inhibitor hit rate was calculated for each kinase. Analysis of the type-II hit rate distributions for STKs and TKs from the Davis assay (*Figure 1A*) indicates that STKs, on average, have an unfavorable contribution to the binding of type-II inhibitors relative to TKs.

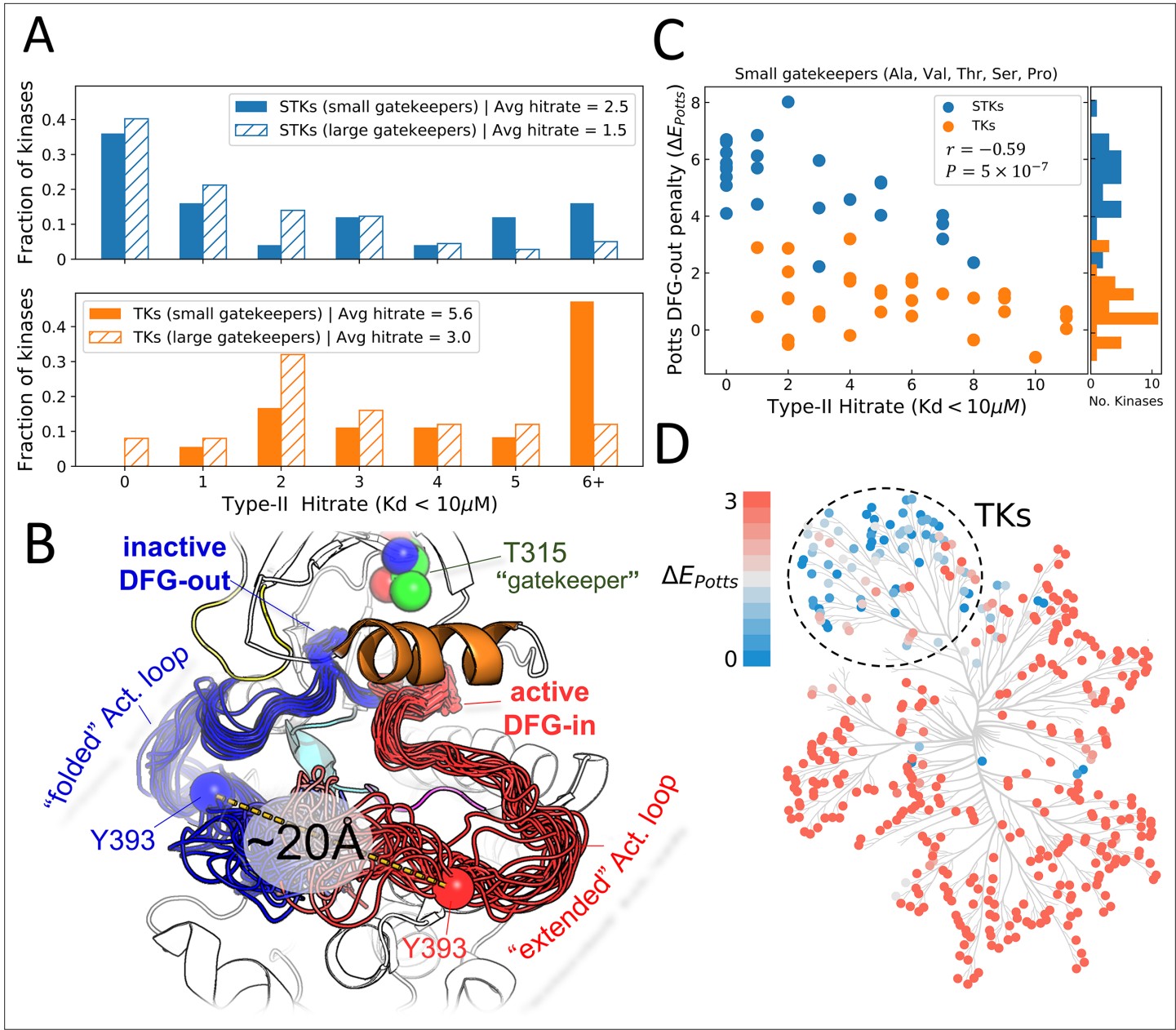

**Figure 1.** Viewing the conformational landscape of the human kinome. (**A**) Hit rate distributions from kinome-wide experimental binding assays with type-II inhibitors for human serine/threonine kinases (STKs; blue, top) and tyrosine kinases (TKs; orange, bottom) with small gatekeepers (solid bars; sidechain volume <110 Å$^3$) and large gatekeepers (hatched bars; sidechain volume >110 Å$^3$). (**B**) PyMol (*pymol, 2015*) visualization of two conformational ensembles populated by Abl kinase from recent solution NMR (Nuclear Magnetic Resonance) experiments (*Xie et al., 2020*). The active DFG-in conformation where the activation loop is 'extended' (red, the Protein Data Bank [PDB]: 6XR6) and the inactive classical DFG-out conformation where the activation loop is 'folded' (blue, PDB: 6XRG) both exist in the absence of ligands, but there is a free-energy cost to transition between them (*Xie et al., 2020*). Type-II inhibitors preferentially bind to this folded DFG-out state. (**C**) Correlation between Potts DFG-out penalty (ΔE Potts) and hit rates for kinases with small gatekeepers only, to control for gatekeeper effects (Pearson correlation of –0.59, p<0.001). (**D**) Potts DFG-out penalties calculated for the human kinome and plotted using CORAL (*Metz et al., 2018*); the TK branch appears to have lower penalties relative to the rest of the kinome, which represent STKs. See *Figure 1—source data 1* for values of the calculated type-II hit rates and Potts threaded-energy scores over the human kinome.

The online version of this article includes the following source data and figure supplement(s) for figure 1:

**Figure supplement 1.** Conformational landscape of the activation loop derived from the Protein Data Bank (PDB).

**Figure supplement 2.** Contact frequency differences between the active DFG-in (BLAminus) and classical DFG-out (BBAminus) conformations.

**Figure supplement 3.** Gatekeeper analysis.

*Figure 1 continued on next page*

Figure 1 continued

**Source data 1.** Type-II inhibitor hit rates and Potts threaded-energy penalties for DFG-out calculated for tyrosine kinases and serine/threonine kinases from the human kinome.

The size of the gatekeeper residue is important for type-II binding as it controls access to a hydrophobic pocket adjacent to the ATP binding site that is traversed by type-II inhibitors (*Zuccotto et al., 2010*; *Bosc et al., 2015*; *Liu et al., 1998*; *Ghose et al., 2008*; *van Linden et al., 2014*), and the size of the gatekeeper residue is thought to negatively affect type-II binding (*Zuccotto et al., 2010*; *Azam et al., 2008*; *Lovera et al., 2015*; *Yun et al., 2008*). Because TKs tend to have small gatekeepers in comparison to STKs (*Zuccotto et al., 2010*; *Taylor and Kornev, 2011*), we considered this as a possible explanation behind the bias for TKs to have larger type-II hit rates. By plotting the hit rate distributions for STKs and TKs where the gatekeeper is either small or large (*Figure 1A*) we confirm that gatekeeper size has an important influence on type-II binding for both STKs and TKs (*Azam et al., 2008*). However, the hit rate distribution for TKs appears more sensitive to gatekeeper size than STKs. Even with small gatekeepers, there is a significant fraction of STKs that have hit rates of zero compared with TKs, suggesting the difference in hit rates between TKs and STKs cannot be accounted for primarily by the size of the gatekeeper residue.

Recent solution NMR experiments with Abl kinase revealed two DFG-out conformational states (*Xie et al., 2020*) one where the DFG motif has flipped 'DFG-in to DFG-out' but the activation loop remains in a 'minimally perturbed' active-like conformation, and the other state is a classical 'folded' DFG-out conformation where the activation loop has moved ~17 Å away from the active conformation (*Figure 1B*), and the DFG motif is in a 'classical DFG-out' (*Vijayan et al., 2015*) or 'BBAminus' (*Modi and Dunbrack, 2019b*) state. Type-II inhibitors were shown to preferentially bind to this folded DFG-out state, confirming observations that Abl is almost always co-crystallized with type-II inhibitors in this conformation. This binding behavior is also exhibited by other kinases, suggested by the large number of activation loop folded DFG-out states seen in type-II bound co-crystal structures (*Figure 1—figure supplement 1*). Hence, the importance of large-scale activation loop conformational changes in type-II binding and the large number of residue-residue contact changes involved in this transition (*Figure 1—figure supplement 2*) suggests the sequence variation of the activation loop and the catalytic loop with which it interacts, might contour the conformational landscape differently for TKs compared with STKs. To investigate this, we used a Potts statistical energy model of sequence covariation to estimate the energetic cost of the active DFG-in (activation loop extended) → inactive DFG-out (activation loop folded) transition for human TKs and STKs (see Methods).

Patterns of coevolution of amino acids at different positions in an MSA are thought to largely reflect fitness constraints for fold stability and function between residues close in 3D space (*Lapedes et al., 2012*; *Hopf et al., 2015*; *Hopf et al., 2017*; *Morcos et al., 2014*), and these coevolutionary interactions can be successfully modeled by a Potts Hamiltonian (*Weigt et al., 2009*; *Lunt et al., 2010*) which we inferred using Mi3-GPU, an algorithm designed to solve 'Inverse Ising' problems for protein sequences with high accuracy (*Haldane and Levy, 2021*). The pairwise interactions from the Potts model can be used as a simple threaded energy function to estimate energetic differences between two conformations, based on changes in residue-residue contacts in the PDB (*Haldane et al., 2016*). We have calculated the threading penalty for all kinases in the human kinome. Our calculations show the Potts predicted DFG-out penalty ($\Delta E_{Potts}$), which is dominated by large-scale reorganization of the activation loop to the folded DFG-out state, is correlated with type-II hit rates (*Figure 1C*) when controlling for gatekeeper size. From this, we determine that sequence variation of the activation loop and the contacts broken/formed by its large-scale conformational change (*Figure 1—figure supplement 2*) makes an important contribution to the binding affinity of type-II inhibitors.

Notably, our calculations over the entire human kinome show that the large majority of kinases with large $\Delta E_{Potts}$ (unfavorable conformational penalties) are STKs, and the large majority of low-penalty kinases are TKs (*Figure 1D*). To validate this finding, we next perform an independent and more computationally intensive prediction of the conformational reorganization energy of TKs and STKs for select kinase targets, chosen based on the kinome calculations of $\Delta E_{Potts}$ and type-II hit rates shown in *Figure 1*, in which we use type-II inhibitors as probes in ABFE simulations as described in the following section. By comparing the conformational penalties predicted from these structure-based molecular dynamics (MD) free-energy simulations with the Potts conformational penalty scores, we

also identify the scale of $\Delta E_{Potts}$ in physical free-energy units. This allows us to predict physical conformational free energies based on Potts calculations which can be carried out at scale on large numbers of sequences, to evaluate the evolutionary divergence of the conformational penalty between STKs and TKs generally.

## Structure-based free-energy simulations guided by the sequence-based Potts model

Relative binding free-energy simulations are now widely employed to screen potent inhibitors in large-scale drug discovery studies (*Wang et al., 2015*). These methods are used to determine the relative free energy of binding between ligands that differ by small substitutions, which permit one to simulate along an alchemical pathway that mutates one ligand to another. By leaving the common core scaffold unperturbed, the cost and difficulty of sampling the transition between unbound (apo) and bound (holo) states of the system are avoided (*Wang et al., 2015*; *Hayes et al., 2022*; *Guest et al., 2022*). Alternatively, alchemical methods to determine ABFEs, such as the 'double decoupling' method employed in this work, sample the apo → holo transition along a pathway that decouples the entire ligand from its environment. While more computationally expensive, the advantage of ABFE is that the computed $\Delta G_{bind}$ can be directly compared with experimental binding affinities, and successful convergence does not rely on the structural similarity of compounds being simulated (*Cournia et al., 2020*; *Li et al., 2020*; *Heinzelmann and Gilson, 2021*; *Lee et al., 2020*; *Gilson et al., 1997*; *Qian et al., 2019*; *Sun et al., 2022*).

Our alchemical ABFE simulations of type-II inhibitors binding to TKs and STKs simulate the apo and holo states of the kinase domains in the classical DFG-out conformation with the activation loop folded, starting from the experimentally determined co-crystal structure of the holo state. The apo state remains DFG-out with the activation loop folded throughout the simulations, and therefore the calculated ABFE ($\Delta G_{bind}^{ABFE}$) excludes the cost to reorganize from DFG-in ($\Delta G_{reorg}$). On the other hand, standard binding free-energies ($\Delta G_{exp}^{o}$) determined experimentally from inhibition or dissociation constants (*Equation 1*) implicitly include the free-energy cost to reorganize. Therefore given the experimentally determined total binding free energy, $\Delta G_{exp}^{o}$ , ABFE simulations can be used to separate the free energy of ligand-receptor association in the inactive state ($\Delta G_{bind}^{ABFE}$) from the cost to reorganize from the active to inactive state, $\Delta G_{reorg}$ (*Equation 3*; *Deng et al., 2011*; *Lin et al., 2014*; *Lin et al., 2013*).

We calculated $\Delta G_{exp}^{o}$ (*Equation 1*) from literature reported $IC_{50}$ or $K_d$ values, where the standard concentration $C_0$ is set to 1 $M$

$$\Delta G_{exp}^{o} = k_b T \, ln \left( K_d / C_0 \right) \tag{1}$$

$\Delta G_{exp}^{o}$ can be expressed as the sum of the free-energy change to reorganize from the active to inactive state, $\Delta G_{reorg}$ plus the free energy to bind to the inactive state $\Delta G_{bind}^{ABFE}$ (*Equation 2*). $\Delta G_{reorg}$ is therefore the excess free-energy difference between $\Delta G_{exp}^{o}$ and $\Delta G_{bind}^{ABFE}$ (*Equation 3*).

$$\Delta G_{exp}^{o} = \Delta G_{reorg} + \Delta G_{bind}^{ABFE} \tag{2}$$

$$\Delta G_{reorg} = \Delta G_{exp}^{o} - \Delta G_{bind}^{ABFE} \tag{3}$$

Type-II inhibitors generally bind when the activation loop is in a folded DFG-out conformation (*Figure 1B*), which presents major challenges for direct simulations to determine the free energy cost of the conformational change in contrast to the method employed here (*Equation 3*).

Because the type-II inhibitor imatinib is co-crystallized in a type-II binding mode with MAPK14 (p38α), an STK, and several other TKs (e.g. ABL1, DDR1, LCK, CSF1R, KIT, and PDGFRA), we chose this inhibitor as an initial probe of our hypothesis that TKs evolved to have lower $\Delta G_{reorg}$ than STKs (*Figure 2*). In this example we note that TKs bind strongly to imatinib ('STI' in *Figure 2*) with an average $\Delta G_{exp}^{o}$ of –9.3 kcal/mol, in contrast to the STK MAPK14 which binds this drug very weakly ($\Delta G_{exp}^{o} = -6.1$ kcal/mol). At face value this appears consistent with our analysis from *Figure 1D*, where we calculated a large Potts DFG-out penalty for MAPK14 ($\Delta E_{Potts} = 5.2$) and low penalties for TKs, suggesting that the weak binding of imatinib to MAPK14 is due at least partially to large $\Delta G_{reorg}$ . To confirm this, we used ABFE simulations with the imatinib: MAPK14 complex to evaluate *Equation 3*,

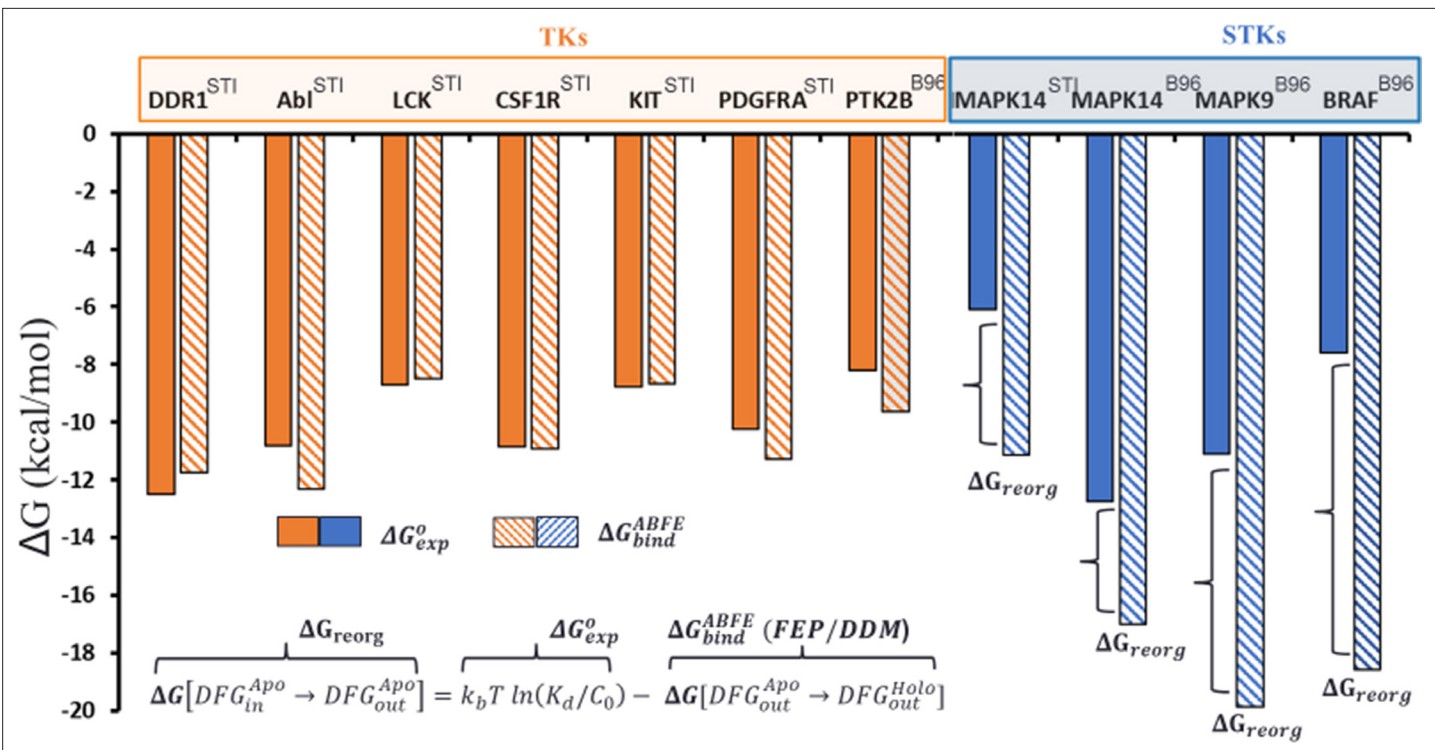

**Figure 2.** Overview of the conformational landscapes between serine/threonine kinases (STKs) and tyrosine kinases (TKs) from absolute binding free-energy simulations, where we compare ΔG$_{bind}$ (hatched bars) from binding free-energy simulations with ΔG$_{exp}$ (solid bars) for the type-II inhibitors imatinib (the Protein Data Bank [PDB] code: STI) and BIRB-976 (PDB code: B96) vs several TKs (orange) and STKs (blue).

The online version of this article includes the following source data and figure supplement(s) for figure 2:

**Figure supplement 1.** Absolute binding free-energy (ABFE) benchmarking results.

**Source data 1.** Absolute binding free-energy results for kinases bound to imatinib and BIRB-796.

confirming that MAPK14 incurs a large penalty to adopt the DFG-out conformation with the activation loop folded ($\Delta G_{reorg}$ = 5 kcal/mol) (**Figure 2**).

Despite the large $\Delta G_{reorg}$ predicted for MAPK14 by both the Potts model and the simulations with imatinib described above, highly potent type-II inhibitors have been successfully developed for this kinase. For example, BIRB-796 (**Pargellis et al., 2002**) binds to MAPK14 about 7 kcal/mol more strongly than imatinib. This stronger binding of BIRB-796 is captured by $\Delta G_{bind}^{ABFE}$ from our simulations (**Figure 2**), and the calculated value of $\Delta G_{reorg}$ for this complex ($\Delta G_{reorg} \approx 4$ kcal/mol) is very close to the corresponding estimate of $\Delta G_{reorg}$ based on simulations with imatinib (**Figure 2**). Importantly, this result suggests that STKs can be potently inhibited by type-II inhibitors despite their large $\Delta G_{reorg}$. To support this, we performed additional ABFE simulations with BIRB-796 and calculated $\Delta G_{reorg}$ for two additional STKs predicted to have large reorganization penalties (MAPK9 and BRAF, $\Delta E_{Potts} \geq 4$). We calculated $\Delta G_{reorg} > 8$ kcal/mol for MAPK9 and BRAF, which is consistent with predictions from the Potts model, and comparison of $\Delta G_{exp}^{o}$ and $\Delta G_{bind}^{ABFE}$ in **Figure 2** confirms that BIRB-796 is able to overcome the large $\Delta G_{reorg}$ of certain kinases to attain high experimental potencies (e.g. MAPK14 and MAPK9). To further validate this result, we calculated $\Delta G_{reorg}$ via ABFE simulations of BIRB-796 binding to a TK predicted by the Potts model to have a low penalty (PTK2B, $\Delta E_{Potts} < 1$), which again shows consistency with our Potts prediction of the conformational landscape (**Figure 2**). The relatively weak value of $\Delta G_{bind}^{ABFE}$ for this kinase compared with MAPK14 is also consistent with observations of the BIRB-796: PTK2B co-crystal structure (PDB: 3FZS), where the binding mode in PTK2B is more weakly associated with the ATP pocket in comparison with MAPK14 (**Han et al., 2009**).

The analysis above provides initial support for our hypothesis about the evolutionarily divergent STK and TK conformational landscapes. To further develop this approach, we identified five STKs and five TKs which are predicted by the Potts threading calculations to have large and small $\Delta G_{reorg}$, respectively, and for which there are sufficient experimental structural and inhibitory data (co-crystal

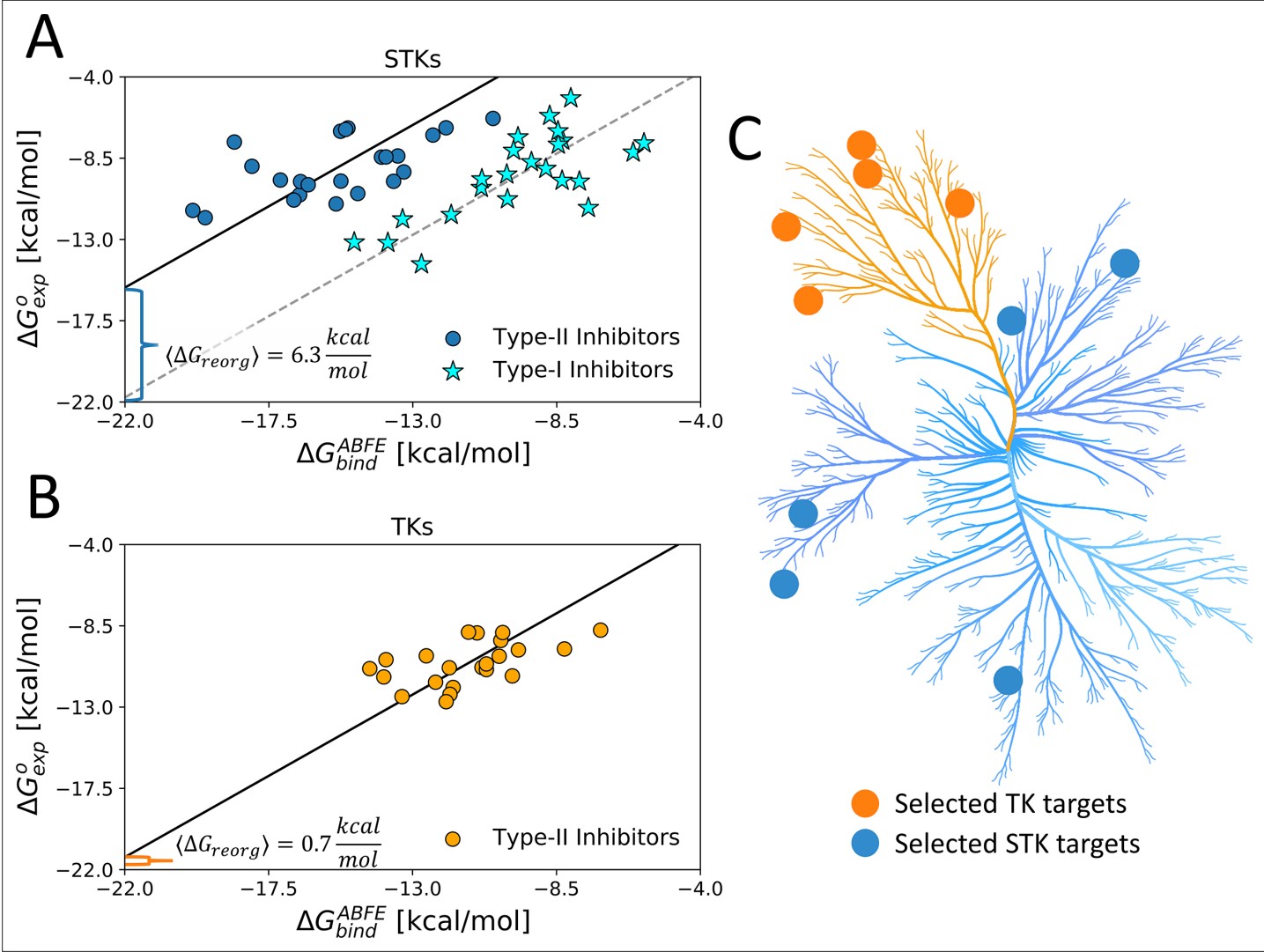

**Figure 3.** Using type-II inhibitors as tools to probe the conformational landscape of TKs and STKs. (**A**) The average $\Delta G_{reorg}$ calculated via absolute binding free-energy (ABFE) simulations with 23 type-I (stars) and 23 type-II inhibitors (circles) complexes in the active DFG-in and inactive DFG-out state, respectively, computed from five serine/threonine kinase (STK) targets (**Table 1**) and (**B**) computed with 22 type-II inhibitors vs five tyrosine kinase (TK) targets in the DFG-out state (**Table 1**). (**C**) Kinome plot created with CORAL (**Metz et al., 2018**) illustrating the selection of five TKs and five STKs which are detailed in **Table 1**.

The online version of this article includes the following source data and figure supplement(s) for figure 3:

**Source data 1.** Absolute binding free-energy results for five tyrosine kinases and five serine/threonine kinases bound to type-II and type-I inhibitors.

**Figure supplement 1.** Experimental binding free-energy vs calculated absolute binding free-energy for type-I inhibitors vs serine/threonine kinases (STKs; column **A**), type-II inhibitors vs STKs (column **B**), and type-II inhibitors vs tyrosine kinases (TKs; column **C**).

structures and binding constants) to calculate an average $\Delta G_{reorg}$ via **Equation 3** for each target using multiple type-II inhibitor probes. For each of the TK and STK targets, these sets of calculations can be visualized as a linear regression of $\Delta G_{exp}^{o}$ vs $\Delta G_{bind}^{ABFE}$ where the slope is constrained to one, consistent with **Equation 2** (see Methods for details). We employed this workflow for the set of five TK and five STK targets by simulating 22 and 23 type-II inhibitor complexes, respectively.

The result of this workflow for the set of five TKs and their type-II complexes revealed a low average $\Delta G_{reorg}$ of <1 kcal/mol (**Figure 3B**), consistent with our initial predictions from Potts $\Delta Es$ and type-II hit rates (**Table 1**). On the other hand, the binding free-energy simulations for the set of five STKs and their type-II complexes show an average of ~6 kcal/mol of $\Delta G_{reorg}$ is required for these kinases to adopt DFG-out conformation, which is also consistent with our initial predictions from the Potts model

**Table 1.** Calculation of reorganization free-energy for five TKs and five STKs.

Type-II hit rates from Davis et al. and Potts threaded energy penalties from *Figure 1* were used to guide the selection of five serine/threonine kinase (STK) and five tyrosine kinase (TK) targets for absolute binding free-energy simulations. For some kinases, the hit rate binary classifier captures a set of relatively weak hits with average binding which, in context with large Potts penalty (see *Figure 1D*), might be explained by a large $\Delta G_{reorg}$ incurred for the folded DFG-out state (*Figure 1B*). See *Figure 3—source data 1* for detailed data and references for experimental binding affinities.

| Kinase | Class | Hit rate[*] ($K_d$ <10 μM) | Potts penalty[†] ($\Delta E_{Potts}$) | Calculated $\Delta G_{reorg}$[‡] |
|--------|-------|------------------|------------------------|---------------------|
| MELK | STK | 3 | 5.9 | 5.6±0.2 |
| MAPK9 | STK | 5 | 4.7 | 6.9±0.3 |
| CDK2 | STK | 2 | 5.3 | 7.7±0.2 |
| IRAK4 | STK | 0 | 2.5 | 5.4±0.2 |
| BRAF | STK | 7 | 4.0 | 6.5±0.1 |
| ABL1 | TK | 10 | −1.0 | 1.3±0.3 |
| LCK | TK | 11 | 0.5 | 1.0±0.3 |
| TIE2 | TK | 6 | 1.1 | −0.3±0.2 |
| NTRK2 | TK | 6 | −1.0 | 1.7±0.1 § |
| DDR1 | TK | 11 | 0.4 | 0.3±0.3 |

[*]Type-II inhibitors only, data from **Davis et al., 2011**.

[†]Calculations from **Figure 1d**.

[‡]$\Delta G_{reorg}$ was calculated from **Equation 3**. Reported standard deviations were calculated by propagating error from the simulations used in the calculation of average $\Delta G_{reorg}$ in units of kcal/mol (see **Figure 3—source data 1** for statistics from individual simulations).

[§]Average and standard deviation calculated from two simulated complexes.

(*Table 1*). To verify that the large $\Delta G_{reorg}$ identified for STKs is a property of conformational selection for DFG-out rather than systematic overestimation of $\Delta G_{bind}^{ABFE}$ for these kinases, we performed ABFE simulations of type-I inhibitors binding to the same set of STKs (an additional 23 complexes). For the binding of type-I inhibitors, we expect there to be no reorganization penalty due to the lack of DFG-out conformational selection in their binding mechanism. As anticipated, the calculated values of $\Delta G_{bind}^{ABFE}$ for type-I inhibitors are very close to their experimental binding affinities ($\Delta G_{exp}^{o}$) (*Figure 3A*).

We find that the set of type-II inhibitors complexed with STKs in this dataset tends to have more favorable binding free energies to the reorganized receptor ($\Delta G_{bind}^{ABFE}$) than type-II inhibitors complexed with TKs, as shown by their distribution along the horizontal axes in *Figure 3*. The reason for this can be understood as a consequence of selection bias. Our selection of STK complexes for this study usually involved lead compounds from the literature, which were designed for high on-target experimental potency and published for their pharmaceutical potential, similar to BIRB-796: MAPK14 which is a tightly bound complex with high experimental affinity despite the large $\Delta G_{reorg}$ incurred by this kinase (*Figure 2*). This tight binding is reflected by the favorability of the $\Delta G_{bind}^{ABFE}$ term which must be implicitly tuned by medicinal chemists to overcome the large $\Delta G_{reorg}$ found in STKs. Meanwhile, the chemical space of type-II inhibitors studied against TKs appears to be privileged by their low $\Delta G_{reorg}$, judging by the comparably weak $\Delta G_{bind}^{ABFE}$ for these complexes. This ultimately gives rise to similar experimental potencies for the binding of type II inhibitors to TKs and STKs plotted in *Figure 3*.

The results of the MD binding free-energy simulations when combined with experimental binding affinities, reveal significant differences in the conformational free-energy landscapes between STKs and TKs. The DFG-in (activation loop extended) to DFG-out (activation loop folded) reorganization penalties are strongly correlated with corresponding $\Delta Es$ calculated from the Potts model ($R^2 = 0.75$, $P \approx 10^{-3}$) emphasizing the connection between coevoluntary statistical energies in sequence space and physical free-energies in protein conformational space (*Figure 4*). From this relationship, we can approximate a scale for the Potts $\Delta E$ scores in physical free-energy units which describe the conformational landscapes of folded proteins in a similar manner to that of an earlier

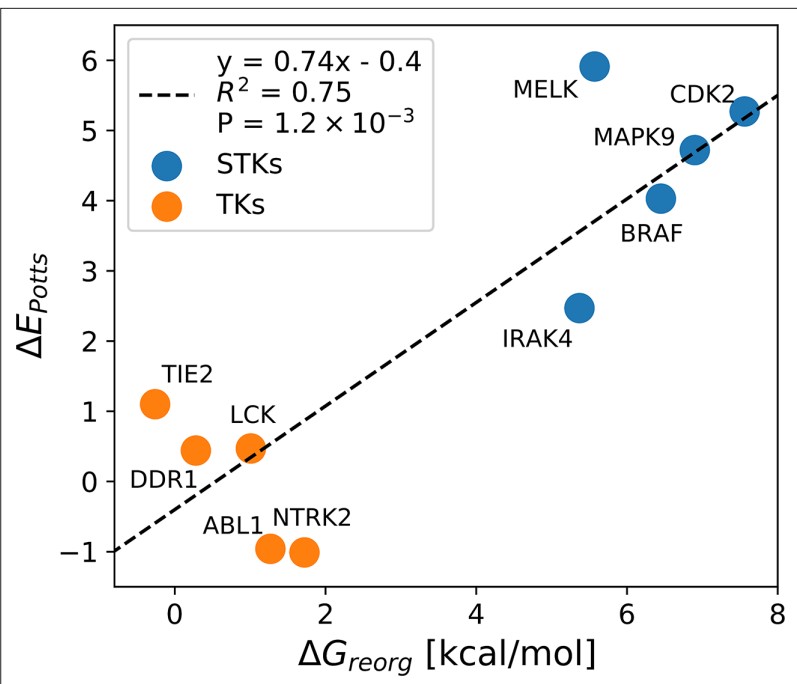

**Figure 4.** Correlation between $\Delta E_{Potts}$ and averaged calculations of $\Delta G_{reorg}$ for five tyrosine kinases (TKs) and five serine/threonine kinases (STKs; *Table 1*).

study of protein folding landscapes (*Morcos et al., 2014*); we find that a Potts statistical energy difference $\Delta E$ of one unit corresponds approximately to 1.3 kcal/mol of $\Delta G_{reorg}$ .

## Structural and evolutionary basis for the divergent TK and STK conformational landscapes

The consistency of the predictions between $\Delta G_{reorg}$ and $\Delta E_{Potts}$ identified from Potts-guided free-energy simulations (*Figure 4*) led us to investigate whether the observed difference in the free energy to reorganize from the active to inactive state is a more general feature that distinguishes TKs from STKs. To this end, we extracted ~200,000 STKs and ~10,000 TKs from the large MSA of Pfam sequences used in the construction of our Potts model, based on patterns of sequence conservation that clearly distinguish the two classes (see Methods). For each sequence, we calculated $\Delta E_{Potts}$ threaded over the structural database (a total of 4268 active DFG-in and 510 classical DFG-out PDB structures) and plotted the distributions for TKs and STKs, revealing a bias for STKs toward larger Potts conformational penalties (*Figure 5*). The average difference between these distributions, $\Delta\Delta E = 3.2$, is extremely unlikely to be observed by chance (p≤$10^{-15}$, see Methods) and supports the hypothesis that TKs are evolutionarily biased toward a lower free-energy cost to adopt the classical 'folded activation loop' DFG-out conformation ($\Delta G_{reorg}$) compared to STKs. We estimate that $\Delta\Delta E = 3.2$ corresponds to ~4.3 kcal/mol based on the analysis summarized in *Figure 4*.

To gain insight into the molecular basis for this effect which distinguishes the conformational landscape of TKs from STKs, we examined the residue-residue interactions that make the most significant contributions to the observed $\Delta\Delta E$. The difference between average Potts threaded-energy penalties, $\Delta\Delta E = \langle\Delta E\rangle_{STKs} - \langle\Delta E\rangle_{TKs}$ , can also be written as a sum over pairs of alignment positions $i$ and $j$ along length $L$ of the aligned kinase domains, $\sum_{|i-j|>4}^{L} \Delta\Delta E_{ij}$ (see Methods for details). We find that ~75% of the total contribution to $\Delta\Delta E$ (approximately 3 kcal/mol) can be traced back to a small number (10) of residue-residue interactions involving the activation loop, suggesting that mutations within the activation loop are largely responsible for the evolutionary divergence between the conformational free-energy landscapes of TKs and STKs. These interactions occur between important structural motifs responsible for controlling the stability of the active 'extended' conformation of the activation loop (*Figure 6A*), especially the activation loop N-terminal and C-terminal 'anchors' (*Nolen*

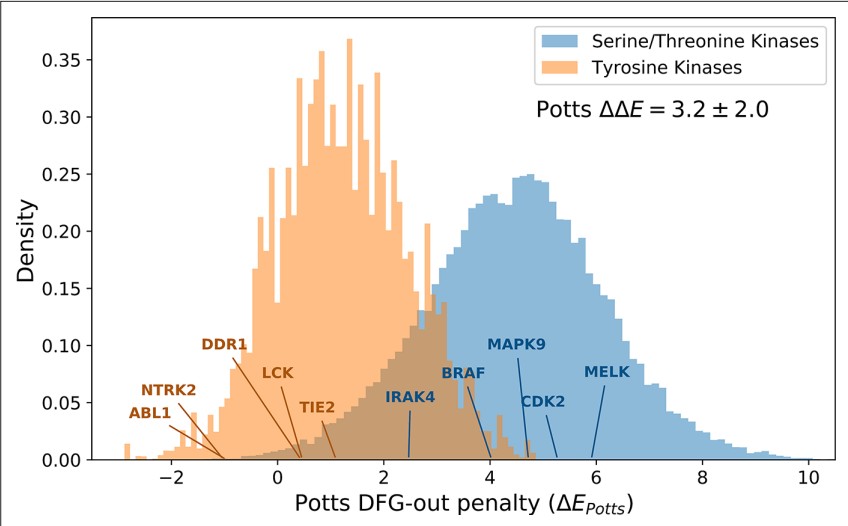

**Figure 5.** The distributions of Potts conformational penalties for (orange) 10,345 tyrosine kinases (TKs) from 471 different species and (blue) 210,862 serine/threonine kinases (STKs) from 2713 different species, showing that TKs tend to have smaller energetic penalties on average. The difference in averages between these distributions is shown ($\Delta\Delta E = 3.2$), which we estimate to be ≈4.3 kcal/mol based on the analysis in **Figure 4**. The probability density was plotted after down-weighting each sequence by the number of times another sequence in the same class (e.g. TKs) is observed within 40% identity (see Methods for details). The effective number of TKs ($N_{eff}$ =1,096) and STKs ($N_{eff}$ =22,893) s the sum totals of their down-weights, which is an unbiased measure of the sequence diversity in each probability distribution.

et al., 2004), and the regulatory 'RD-pocket' formed by the HRD motif of the catalytic loop which functions to stabilize or destabilize this conformation depending on the activation loop's phosphorylation state (**Nolen et al., 2004**; **Johnson et al., 1996**). The remaining top $\Delta\Delta E_{ij}$ correspond to contacts that stabilize the DFG-out 'folded' conformation of the activation loop for TKs, wherein the kinase's substrate binding site recognizes its own activation loop tyrosine (**Figure 6C**, right; **Hubbard et al., 1994**). We describe these interactions below, focusing on the strongest effects involving these structural motifs that lead to differences in the conformational free-energy landscapes of TKs and STKs. The residue nomenclature we use in our descriptions follows the format $MSA_{ABL1}^{CDK6}$, which is the unique residue numbering in our MSA followed by Abl1 (TK) numbering in the subscript (active PDB: 2GQG, inactive PDB: 1IEP) and CDK6 (STK) numbering in the superscript (active PDB: 1XO2, inactive PDB: 1G3N), corresponding to the original PDB files used to generate **Figure 6B** and **Figure 6C**.

The 'RD-pocket' (**Nolen et al., 2004**) is a conserved basic pocket formed by the Arg and Asp residues of the HRD motif (**Johnson et al., 1996**) ($R123_{361}^{144}$ and $D124_{362}^{145}$) and a positively charged Lys or Arg that is often present in the N-terminal anchor of the activation loop ($147_{386}^{168}$). $R123_{361}^{144}$ in the HRD motif and Lys/Arg $147_{386}^{168}$ in the N-terminal anchor form an unfavorable like-charge interaction when the activation loop is in the active, extended conformation (**Nolen et al., 2004**). Kinase activation is typically a complex process involving many layers of regulation from other protein domains, cofactors, and phosphorylation events (**Endicott et al., 2012**) however, a general activation mechanism that applies to the majority of kinases involves quenching the net-charge of the RD-pocket by addition of a negatively charged phosphate group to a nearby residue in the activation loop, stabilizing the active conformation. The conservation of this regulatory mechanism in most protein kinases, particularly those bearing the HRD-Arg residue (termed 'RD-kinases' **Johnson et al., 1996**), explains why Lys or Arg is frequently observed at position $147_{386}^{168}$ of the N-terminal anchor of the activation loop. However, RD-TKs prefer Arg at this position (78%) which the Potts model suggests has a greater destabilizing effect on the active conformation than Lys (9%) due to interactions with HRD-Arg. The activation loop Arg also forms part of an electrostatic interaction network that stabilizes the 'Src-like inactive' conformation in TKs (**Ozkirimli and Post, 2006**; **Wu et al., 2020**), a conformation with a 'partially' folded activation loop (**Figure 1—figure supplement 1**) that is suggested to be an intermediate state along the transition to DFG-out (**Levinson et al., 2006**; **Shan et al., 2013**). On the other hand, RD-STKs

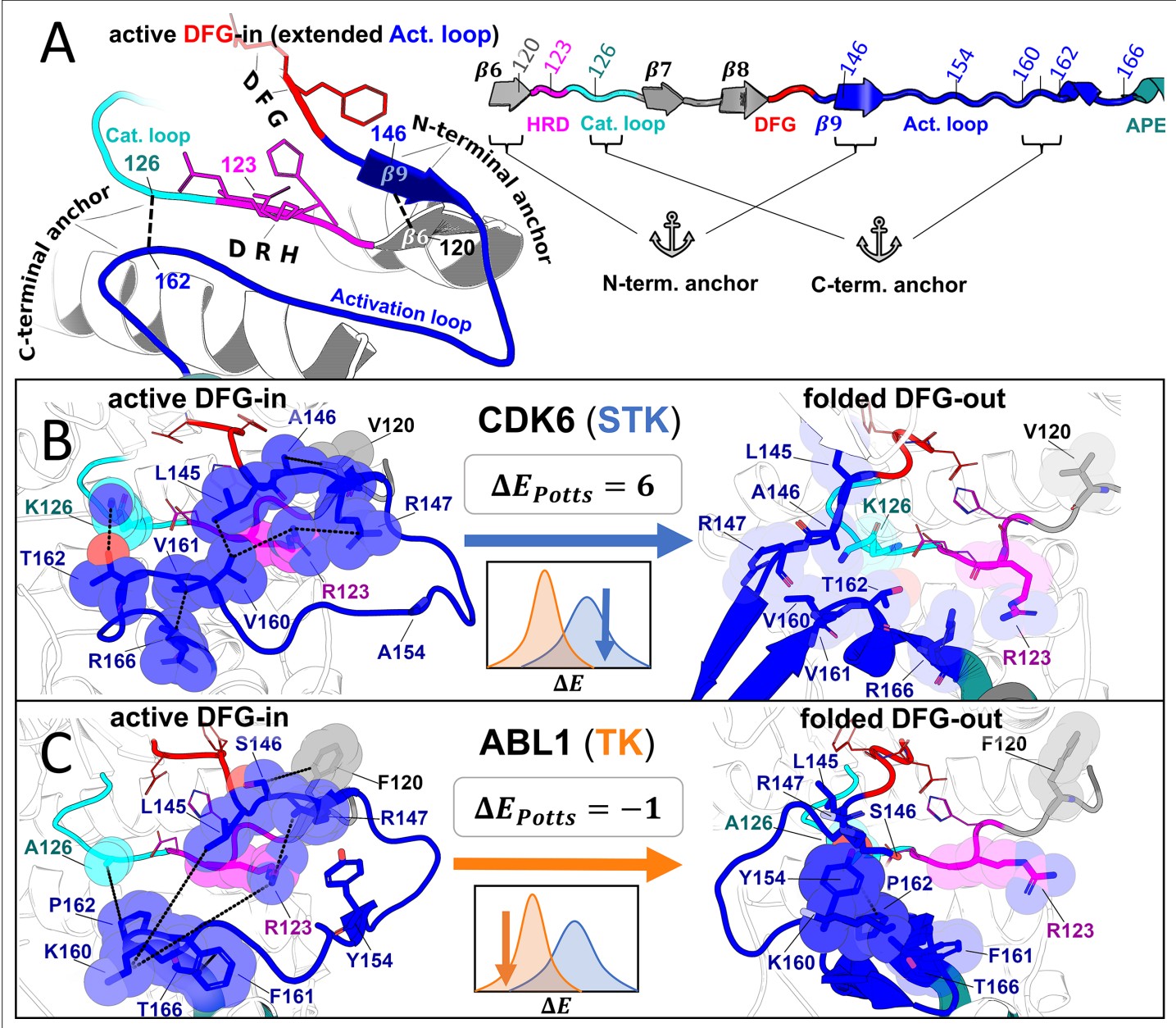

**Figure 6.** Molecular basis for the evolutionary divergence between tyrosine kinases (TKs) and serine/threonine kinases (STKs). Visualizing the top interaction pairs that contribute to the result in **Figure 5**, which is discussed in the main text. (**A**) Diagrams depicting general features of the active 'extended' conformation of the activation loop (left) and the primary structure of these motifs in our multiple sequence alignment (MSA; right) with the HRD (pink), DFG (red), and APE (teal) motifs color-coded for reference. (**B–C**) Structural examples of a representative STK (CDK6) and TK (Abl) in the active DFG-in conformation (left) and the folded DFG-out conformation (right). Residues are labeled according to their position in our MSA, and colored according to A. The inset (center) displays the ΔE$_{Potts}$ of the reference kinase derived from **Figure 5** as well as a cartoon depicting their location in the distributions. The diagrams of CDK6 in the active DFG-in conformation (PDB: 1XO2, chain B), CDK6 in the folded DFG-out conformation (PDB: 1G3N, chain A), Abl in the active DFG-in conformation (PDB: 2G2I, chain A), and Abl in the folded DFG-out conformation (PDB: 1IEP, chain A) were generated with PyMol (**pymol, 2015**). All ligands and some backbone atoms were hidden for clarity.

The online version of this article includes the following figure supplement(s) for figure 6:

**Figure supplement 1.** Interactions belonging to the top five contributions to ΔΔE.

**Figure supplement 2.** Cumulative sum of position pair contributions to ΔΔE.

display $K147$ more frequently (26%), which the Potts model suggests interacts more favorably with the HRD-Arg independently of activation loop phosphorylation, thus contributing to greater stabilization of the active DFG-in conformation for RD-STKs in comparison with TKs. Additionally, the Potts statistical energy analysis suggests that packing interactions between HRD-Arg and $V160_{400}^{180}$ located near the activation loop C-terminal also contributes to phosphorylation-independent stabilization of the active conformation in RD-STKs (*Figure 6B left*), appearing in 28% of RD-STKs and only 2% of RD-TKs. In RD-TKs, the residue at this position is usually Arg or Lys which appears to be repelled from the RD-pocket (*Figure 6C left*) and is suggested by the Potts model to result in a less stable active conformation.

The largest $\Delta\Delta E_{ij}$ term, which contributes 16% of the total difference in average Potts conformational penalties between STKs and TKs, comes from an interaction pair in the C-terminal of the activation loop ($162_{402}^{182}$) and the C-terminal of the catalytic loop ($126_{364}^{147}$). These residues form part of the 'C-terminal anchor' (*Nolen et al., 2004*) which is important for creating a suitable binding site for the substrate peptide. The C-terminal anchor residue $162_{402}^{182}$ is Pro in TKs and typically Ser or Thr in STKs (*Taylor et al., 1995*). In STKs, the sidechain hydroxyl of this residue forms a hydrogen bond with $K126_{364}^{147}$ in the catalytic loop, creating a stable binding site for substrate phosphoacceptor residues and stabilizing the C-terminal anchor (*Taylor et al., 1995*). $K126$ is also directly involved in catalysis by interacting with and stabilizing the gamma phosphate of ATP (*Zheng et al., 1993*); hence, it is often referred to as the 'catalytic lysine'. The hydrogen bond between $K126$ and $S$ or $T162$ is almost always formed in the active DFG-in conformation, and we observe breakage of this hydrogen bond in many STKs crystallized in the DFG-out/activation loop folded conformation (e.g. CDK6, *Figure 6B*), suggesting that deformation of the C-terminal anchor contributes an energetic penalty for the active → inactive conformational change. In TKs, however, the catalytic lysine is almost always replaced with Ala, with the exception of a few TKs (e.g. c-Src) which have instead adopted Arg at this position. The C-terminal anchor of TKs containing $A126$ and $P162$ is less stable in comparison to STKs containing ($K126$, $T162$) or ($K126$, $S162$) for which the Potts coupling is very favorable, consistent with the structural observation that ($A126$, $P162$) forms weak interactions (*Figure 6C*). Our analysis suggests the interaction pair ($A126$, $P162$) weakens the C-terminal anchor, leading to a less stable active conformation in TKs as compared with STKs. Another significant contribution to the stabilization of the C-terminal anchor in the active DFG-in conformation for STKs comes from interactions between the residue pair ($161_{401}^{181}$, $166_{406}^{186}$) which are both located within the activation loop. We observe ($G161$, $M166$) at this position pair in 33% of STKs, but never in TKs (*Figure 6—figure supplement 1D*). The Potts coupling between these residues is highly favorable. In contrast, we observe ($L161$, $M166$) in 40% of TKs but never in STKs (*Figure 6—figure supplement 1H*), which have weaker coupling. The bulky sidechains of ($L161$, $M166$) observed in TKs cause the activation loop to 'bulge' in this C-terminal region which has been previously identified as a feature of TKs that helps shape the substrate binding site to accommodate Tyr residues (*Nolen et al., 2004*). In addition to this paradigm, our analysis suggests that the C-terminal bulge results in weaker structural constraints on the active conformation relative to STKs.

In summary, TKs are suggested by the Potts statistical energy model which is based on sequence covariation, to have on average, weaker N-terminal anchor, RD-pocket, and C-terminal anchor interactions than STKs. This mechanism of shifting the TK conformational equilibrium away from the active DFG-in/extended activation loop conformation can explain 7 of the top 10 $\Delta\Delta E_{ij}$, accounting for ~80% of these contributions to the divergence between the STK and TK conformational landscapes. The remaining ~20% of the top contributions can be attributed to residue-residue interactions that occur within the folded DFG-out conformation wherein the activation loop of TKs binds to the kinase's own active site as though it was engaging a peptide substrate in trans (*Figure 6C right*) (*Nolen et al., 2004*). STKs, however, rarely adopt a folded DFG-out conformation with this property, and instead the activation loop is typically found to be unresolved and/or projecting outward toward solvent (*Figure 6B right*). The Potts model suggests that this substrate mimicry of the folded DFG-out activation loop observed in TKs is highly dependent on the presence of a Tyr phosphorylation site at position in the activation loop (*Figure 6C*). In the active conformation, the anionic p$Y154$ stabilizes the active conformation by binding to the basic RD-pocket (*Figure 6C left*, phosphate not shown). However, in the (unphosphorylated) folded DFG-out conformation, this $Y154$ mimics a substrate by stacking against the TK-conserved $P162_{402}^{182}$ residue (*Figure 6C right*) (*Nolen et al., 2004*). The substrate

mimicking nature of this binding mode is demonstrated by the autophosphorylation dimer structure of TK FGFR3 (PDB: 6PNX) solved recently (*Figure 6—figure supplement 1K*; *Chen et al., 2020*).

The striking connection between the ability of TKs to phosphorylate tyrosine substrates and their enhanced access to the DFG-out conformation via substrate-competitive contacts from their own activation loop described above suggests an evolutionary model for the TK conformational behavior characterized in this work. In this model, the coevolution of residues that form substrate-competitive contacts in folded DFG-out appears to be a byproduct of the evolutionary pressure for TKs to phosphorylate other TKs on their activation loop tyrosine residues. STKs, on the other hand, have optimized the binding of Ser and Thr substrates via a different binding mode (*Endicott et al., 2012*) which does not have the same energetic feedback with the stability of the folded DFG-out conformation. Additionally, the catalytic domains of TKs appear to have a less energetically stable 'extended' activation loop conformation than STKs, which may have encouraged the evolution of more complex mechanisms of allosteric regulation and autophosphorylation which are highly important regulatory mechanisms in TKs (*Chen et al., 2020*; *Beenstock et al., 2016*; *Lemmon and Schlessinger, 2010*). The combined effect of these two TK phenotypes, the former favoring the stabilization of folded DFG-out and the latter favoring *de*stabilization of active DFG-in, may explain their low free-energy cost for the DFG-in → DFG-out conformational change compared with STKs.

## Discussion

In this work, we have combined sequence and structure-based approaches to analyze the conformational free energy difference between active DFG-in and inactive DFG-out kinase states. Using a Potts statistical energy model derived from residue-residue covariation in a kinase family multiple sequence alignment, we first threaded all human STKs and TKs onto large ensembles of active DFG-in and classical DFG-out structures from the PDB. We found distinctly different distributions of threading scores for STKs compared with TKs, with STKs having a significant conformational reorganization penalty compared with TKs. The molecular basis for the evolutionary divergence in the conformational landscapes was analyzed; a substantial contribution to the difference is associated with sequence position pairs that couple the N and C terminal anchor residues of the activation loop to N and C terminal residues in the catalytic loop, according to the Potts statistical energy analysis. We then used the Potts statistical energy model to guide the selection for structure-based MD binding free energy simulations of 74 protein-ligand complexes; using the calculated binding free-energy estimates together with experimental values, we were able to estimate free-energy costs for the large-scale (~17–20 Å) conformational change of the activation loop by an indirect approach. The structure-based estimates of the reorganization free-energy penalties are consistent with the sequence-based estimates. Additionally, the strong correlation between $\Delta G_{reorg}$ and $\Delta E_{Potts}$ identified in this study reveals that the conformational landscape has a strong sequence dependence with STKs having an ~4 kcal/mol conformational free energy bias favoring the active state over the inactive state relative to TKs (*Figure 4*). We note that the most potent type-II inhibitors from the literature which target STKs bind with nanomolar $K_d$s, similar to that for TKs, despite the substantial additional reorganization penalty that STKs must overcome. This suggests that medicinal chemists have implicitly been able to exploit particularly favorable characteristics of the type-II binding pocket to design inhibitors with extremely strong affinities to the DFG-out (activation loop folded) receptor conformations of STKs, and that further analysis of the molecular basis for this tight binding could provide a basis for designing more selective inhibitors.

## Materials and methods
### Multiple sequence alignment (MSA) and classification of serine/threonine vs tyrosine kinases

An MSA of 236,572 protein kinase catalytic domains with 259 columns was constructed as previously described (*McGee et al., 2021*). STKs and TKs were classified based on patterns of sequence conservation previously identified by Taylor and co-workers (*Taylor et al., 1995*); characteristic sequence features of TKs and STKs which form their respective phosphoacceptor binding pockets are found at the HRD +2 (Ala or Arg in TKs, Lys in STKs) and HRD +4 (Arg in TKs, variable in STKs) in the catalytic loop, as well as the APE-2 residue (Trp in TKs, variable in STKs) in the activation loop. These

residues correspond to positions 126, 128, and 165 in our MSA, respectively. Kinases which satisfy the conditions for TKs at all three positions were classified as TKs (10,345 raw sequences, 1069 effective sequences), and those that satisfy the condition for STKs at position 126 and are non-overlapping with the TK condition at position 128 were classified as STKs (210,862 raw sequences, 22,893 effective sequences). The effective number of sequences in each class was calculated by summing over sequence weights, where each sequence was assigned a weight defined as the fraction of the number of sequences in the same class that are within 40% identity. In this way, we correct for the effects of phylogenetics in the calculation of sample size as well as other quantities (see below).

## Human kinome dataset

497 human kinase catalytic domain sequences were acquired from *Modi and Dunbrack, 2019b* (excluding atypical kinases). These sequences were aligned using a Hidden Markov Model (HMM) that contains 259 columns (L=259), which was constructed from the same MSA used to derive our Potts model. 447 human kinases remained after filtering-out sequences with 32 or more gaps.

## Classification of gatekeeper size

The designation of gatekeeper residues as 'large' or 'small' was based on sidechain van der Waals volumes (*Miller et al., 1987*; *Figure 1—figure supplement 3*), where small gatekeepers have a volume of <110 Å$^3$ (Gly, Ala, Ser, Pro, Thr, Cys, Val), and large gatekeepers have a volume of >110 Å$^3$ (Asn, His, Ile, Leu, Met, Lys, Phe, Glu, Tyr, Gln, Trp, Arg).

## PDB dataset and conformational states

X-ray crystal structures of tyrosine, serine/threonine, and dual-specificity eukaryotic protein kinases in the PDB were collected from http://rcsb.org on July 30, 2020. The protein sequences of 7919 chains were extracted from 6805 PDB files by parsing the SEQRES record and aligned to the MSA used to construct our Potts model, using an HMM.

Contact frequency differences between ensembles of active/DFG-in and inactive/DFG-out (classical DFG-out) PDB structures (*Figure 1—figure supplement 2*) are incorporated into the calculation of Potts threaded energies, which are central to this work. Our classification of the active DFG-in and classical DFG-out conformational states is based on ref 6, which we describe in further detail here:

### Active DFG-in (BLAminus)

The BLAminus state of the DFG motif (active DFG-in) is the only active conformation of the activation loop, in contrast to several other inactive DFG-in states (ABAminus, BLAplus, BLBplus, BLBminus, BLBtrans). In the active conformation, all structural requirements for catalytic activity are typically met, e.g., a complete hydrophobic spine, a salt bridge between β3-Lys → αC-Glu, and an extended activation loop that ensures unobstructed substrate-binding, all of which have high correspondence with the BLAminus state (*Modi and Dunbrack, 2019b*). Our analysis using the software provided in *Modi and Dunbrack, 2019b* identifies 3643 structures in this conformation belonging to STKs and 625 structures belonging to TKs (4268 structures in total).

### Classical DFG-out (BBAminus)

The DFG-out state is characterized by a 'flip' of the conserved Phe to occupy the ATP binding pocket which is otherwise occupied by the conserved Asp in the active conformation. The classical DFG-out conformation is associated with the binding of type-II inhibitors which occupy the back pocket region opened up by the DFG-flip (*Vijayan et al., 2015*). This conformation corresponds to the BBAminus rotamer state of these residues, and it is the dominant DFG-out conformation associated with the binding mode of type-II inhibitors. This classical DFG-out or BBAminus conformation is correlated with a larger-scale conformational change of the activation loop that involves an ~17 Å 'folding' transition with respect to the active conformation. This conformational change is usually accompanied by the formation of secondary structure that obstructs the typical substrate binding surface (e.g. PDB ID: 2HIW, *Figure 1—figure supplement 1B*). Our conformational analysis via *Modi and Dunbrack, 2019b* identifies 224 structures in this conformational state belonging to STKs and 286 structures belonging to TKs (510 structures in total).

## Contact frequency differences

Each of the clustered PDB structures were converted to an adjacency matrix of binary contacts (1 for 'in-contact', 0 otherwise). A contact between residues $i$ and $j$ in structure $n$ was assigned when their nearest sidechain atoms (excluding hydrogen) were detected within a distance $r_{ij}(n) < 6\,\mathring{A}$. The contact frequency $c_{ij}^A$ in cluster $A$ (e.g. active DFG-in) was calculated for each residue pair $(i, j)$ by taking a weighted average over all instances of a contact in that cluster –

$$c_{ij}^A = \frac{1}{\sum\limits_n^{N_A} w_n^{AB}} \sum_n^{N_A} w_n^{AB} \delta_{ij}(n) \tag{4}$$

$$\delta_{ij}(n) = \begin{cases} 1, & r_{ij}(n) < 6\,\mathring{A} \\ 0, & r_{ij}(n) \geq 6\,\mathring{A} \end{cases} \tag{5}$$

Where $N_A$ is the number of PDB chains in cluster $A$, and weights were calculated with $w_n^{AB} = \frac{1}{u_n^{AB}}$, where $u_n^{AB}$ is the number of times the UniProt ID of structure $n$ is found within either cluster $A$ (i.e. active) or cluster $B$ (i.e. DFG-out). In this way, we have down-weighted contributions to the contact differences $\Delta c_{AB}^{ij} = c_A^{ij} - c_B^{ij}$ that are due to overrepresentation of specific kinases in the PDB clusters, with the goal of using contact differences to represent conserved features of the conformational transition across many different kinases. Alignment gaps and unresolved residues were accounted for by excluding these counts in the summations. Only $|i - j| > 4$ were included in the calculation. The PDB clusters used to calculate these contact differences are described above. The contact frequency differences for both STKs and TKs were plotted on a contact map for visualization (**Figure 1—figure supplement 2**).

## Potts model and threaded-energy calculation

Our Potts Hamiltonian was constructed from an MSA of protein kinase catalytic domains as previously described (**McGee et al., 2021**). The Potts Hamiltonian $H(S)$ takes the form –

$$H(S) = \sum_{i<j}^L J_{S_i S_j}^{ij} + \sum_i^L h_{S_i}^i \tag{6}$$

where $L$ is the number of columns in the MSA ($L = 259$), $h$ is a matrix of self-interactions or 'fields', and $J$ is the coupling matrix which has the interpretation of co-evolutionary interactions between residues.

The Potts threaded-energy penalty $\Delta E(S)$ for sequence $S$ to undergo the conformational transition $A \rightarrow B$ is calculated using contact frequency differences between the two conformational ensembles (**Haldane et al., 2016**) –

$$\Delta E(S \in X) = -\sum_{i<j}^L J_{S_i S_j}^{ij} \Delta c_{AB}^{ij}(X) \; , \tag{7}$$

where $X$ represents a class or family of sequences for which sequence $S$ has membership, and $\Delta c_{AB}^{ij}(X)$ represents the contact frequency difference between conformations $A$ and $B$ observed only for other sequences belonging to class $X$ (e.g. $X \equiv TKs$ or $X \equiv STKs$; upper and lower triangle of **Figure 1—figure supplement 2**, respectively). As described previously (**Haldane et al., 2016**) the couplings ($J_{S_i S_j}^{ij}$) and fields ($h_{S_i}^i$) were transformed to the 'zero-gauge' prior to calculating $\Delta E(S)$.

Contributions to average shift in $\mathbf{\Delta E_{Potts}}$ between STKs and TKs ($\mathbf{\Delta\Delta E}$). Where $\Delta E(S)$ is the Potts conformational penalty for sequence S to undergo the conformational change $A \rightarrow B$, we define $\Delta\Delta E$ as the difference in average $\Delta E$ between two groups of sequences $X$ and $Y$

$$\Delta\Delta E = \langle \Delta E \rangle_X - \langle \Delta E \rangle_Y \; . \tag{8}$$

To help interpret $\Delta\Delta E$ in a structural and coevolutionary context, we can write $\Delta\Delta E$ as a sum over position pairs $(i, j)$

$$\Delta\Delta E = \sum_{i<j}^{L} \Delta\Delta E^{ij} . \tag{9}$$

To evaluate this, we note that average $\Delta E$ can be expressed as a sum over position pairs

$$\langle \Delta E \rangle_X = \sum_{i<j}^{L} \left\langle \Delta E^{ij} \right\rangle_X \tag{10}$$

where $\left\langle \Delta E^{ij} \right\rangle$ is calculated as follows

$$\left\langle \Delta E^{ij} \right\rangle_X = -\sum_\alpha\sum_\beta f_{\alpha\beta}^{ij}(X) J_{\alpha\beta}^{ij} \Delta c_{AB}^{ij}(X) . \tag{11}$$

$f_{\alpha\beta}^{ij}(X)$ is the frequency (bivariate marginal) of residues $\alpha$ and $\beta$ at positions $i$ and $j$ for sequences in the MSA which belong to group $X$, which we calculate after applying the MSA-derived phylogenetic weights described above. Finally, by substituting *Equation 10* back into *Equation 8*, we show how $\Delta\Delta E$ can be decomposed into contributions from individual residue pairs

$$\Delta\Delta E^{ij} = -\sum_\alpha\sum_\beta J_{\alpha\beta}^{ij} \left( f_{\alpha\beta}^{ij}(X) \Delta c_{AB}^{ij}(X) - f_{\alpha\beta}^{ij}(Y) \Delta c_{AB}^{ij}(Y) \right) . \tag{12}$$

By viewing the largest (most positive) $\Delta\Delta E^{ij}$ terms, where $X \equiv STKs$ and $Y \equiv TKs$ in *Equation 12*, we are identifying position pairs that cause STKs to have higher penalties than TK in our Potts threading calculations for the active DFG-in to DFG-out conformational change (*Figure 6—figure supplement 2*).

## Calculation of p-value for $\Delta\Delta E$

The quantity $\Delta\Delta E$ is a difference between two averages, $\langle \Delta E \rangle_{STK} - \langle \Delta E \rangle_{TK}$ . Hypothesis testing to determine the statistical significance of this quantity was performed with respect to a null model where the populations of $\Delta Es$ for STKs and TKs, from which our samples were drawn, are indistinguishable. To this end, a p-value was calculated for a t-statistic derived from Welch's t-test (*Welch, 1947*), where $s$ is the standard error of average $\Delta E$ –

$$\widetilde{t} = \frac{\langle \Delta E \rangle_{STK} - \langle \Delta E \rangle_{TK}}{\sqrt{s_{STK}^2 + s_{TK}^2}} \tag{13}$$

where the averages and standard errors are calculated after down-weighting each sequence as described above. This was done to lessen the effects of phylogenetic sampling bias from our MSA and ensure that $\Delta\Delta E$ captures general differences between TKs and STKs, rather than specific TK or STK families.

The one-tailed p-value was calculated using the cumulative t-distribution function generated in python using the SciPy package (*Virtanen et al., 2020*),

$$p = 1 - t_{cdf}\left(\widetilde{t}, \nu\right) \tag{14}$$

where the degrees of freedom for the t-distribution describing the combined population, $\nu$, was estimated via the Welch-Satterthwaite equation (*Welch, 1947*) from the degrees of freedom of the two samples $\nu_{STK}$ and $\nu_{TK}$

$$\nu = \frac{\left(s_{STK}^2 N_{STK}^{-1} + s_{TK}^2 N_{TK}^{-1}\right)^2}{s_{STK}^4 N_{STK}^{-2} \nu_{STK}^{-1} + s_{TK}^4 N_{TK}^{-2} \nu_{TK}} . \tag{15}$$

where $N$ represents the effective number of STKs or TKs, which is an unbiased count of sequences in each dataset that can be obtained by summing the sample weights ($N_{STK} = 22,893$, $N_{TK} = 1069$). From the calculation of $\Delta\Delta E = \langle \Delta E \rangle_{STK} - \langle \Delta E \rangle_{TK} = 3.2$, we determine the corresponding p-value to be less than $10^{-15}$, meaning it is highly unlikely for this large of a difference to be observed if the $\Delta Es$ for TKs and STKs were randomly drawn from the same distribution rather than distinct distributions.

## Enumeration of absolute binding free-energy simulations

In this work, we have performed all-atom MD simulations in explicit solvent for a total of 94 different kinase-inhibitor complexes to calculate ABFEs via the alchemical DDM. 74 of these free-energy calculations were guided by insights from the Potts model, specifically the patterns of Potts conformational penalties plotted in *Figure 1D*. 68 of these complexes correspond to the ABFEs plotted in *Figure 3*, of which 23 type-II inhibitors and 23 type-I inhibitors ABFEs are plotted for STKs in *Figure 3A*, and 22 type-II inhibitors ABFEs for TKs are plotted in *Figure 3B*. Six additional Potts-guided ABFEs corresponding to CSF1R, KIT, PDGFRA, MAPK14, and PTK2B are included in *Figure 2*. An additional 20 type-I and type-I ½ ABFEs were calculated as part of our benchmarking procedure described in the Methods section.

## Double decoupling method setup

The double decoupling method (DDM), also known as an 'alchemical' method, was applied to compute ABFE ($\Delta G^o_{bind}$), as shown in *Equation 16* (*Deng et al., 2018*; *Sakae et al., 2020*). This method computes the free energies of decoupling the inhibitor from the bulk solvent in the presence and absence of a receptor via a nonphysical thermodynamic cycle where the two end states are connected via the alchemical pathway. The starting holo-structures for ABFE calculations were taken from the available crystal structure. The absence of crystal structure prompted us to model the structure of the ligand into the active site of the kinase by superimposing over the binding pose of the available holo crystal structure.

$$\Delta G^o_{bind} = -\Delta G^{bound}_{restrain} - \Delta G^{bound}_{decouple} + \Delta G^{gas}_{restrain} + \Delta G^{bulk}_{decouple} \tag{16}$$

Decoupling of the ligand was achieved by first turning off the coulombic intermolecular interactions followed by Lennard-Jones intermolecular interactions from both the legs. This allows DDM to estimate the free energy, i.e., in the presence of protein ($\Delta G^{bound}_{decouple}$) and absence of protein, i.e., in the bulk solvent, ($\Delta G^{bulk}_{decouple}$) as shown in *Equations 17; 18*.

$$\Delta G^{bound}_{decouple} = \Delta G^{bound}_{decouple-Coulomb} + \Delta G^{bound}_{decouple-LJ} \tag{17}$$

$$\Delta G^{bulk}_{decouple} = \Delta G^{bulk}_{decouple-Coulomb} + \Delta G^{bulk}_{decouple-LJ} \tag{18}$$

Substituting *Equations 17; 18* into *Equation 19* yields the estimated ($\Delta G^o_{bind}$) from DDM

$$\Delta G^o_{bind} = -\Delta G^{bound}_{restrain} + \Delta\Delta G_{Coulomb} + \Delta\Delta G_{LJ} + \Delta G^{gas}_{restrain} \tag{19}$$

where $\Delta\Delta G_{Coulomb}$ is the electrostatic energy contribution toward the total ABFE, and $\Delta\Delta G_{LJ}$ is the non-polar energy contribution.

In this study, depending on the system's convergence, either 20 or 31 total $\lambda$s were used for decoupling the ligand from bulk solvent. For instance, either 5 $\lambda$s with $\Delta\lambda=0.5$ or 11 $\lambda$s with $\Delta\lambda=0.1$ were used for coulombic decoupling and 15 $\lambda$s with $\Delta\lambda=0.1$ or 20 $\lambda$s with $\Delta\lambda=0.05$ were used for decoupling Lennard-Jones interactions in the bulk solvent.

Similarly, depending on the convergence, either 30 or 42 total $\lambda$s were used for decoupling ligand bound to protein. For instance, 11 or 12 non-uniformly distributed $\lambda$s were used to restrain the ligand. Decoupling the coulombic interactions between ligand and protein was achieved by either using 4 $\lambda$s with $\Delta\lambda=0.25$ or 10 $\lambda$s with $\Delta\lambda=0.1$, whereas a large number of $\lambda$s were used for decoupling Lennard-Jones interactions, i.e., 15 $\lambda$s with $\Delta\lambda=0.1$ or 20 $\lambda$s with $\Delta\lambda=0.05$ were used. The correction term developed by Rocklin and coworkers for treating charged ligands during DDM simulations was adopted (*Rocklin et al., 2013*). In this regard, it is well documented that the use of a finite-sized periodic solvent box during DDM simulations can lead to non-negligible electrostatic energy contribution toward the calculated total ABFE. Thus, calculated ($\Delta G^o_{bind}$) for charged ligand after addition of electrostatic correction term can be expressed as:

$$\Delta G^o_{bind} = -\Delta G^{bound}_{restrain} + \Delta\Delta G_{Coulomb} + \Delta\Delta G_{LJ} + \Delta G^{finite\_size}_{electrostatic\_correction} + \Delta G^{gas}_{restrain} \tag{20}$$

For a proper convergence during DDM simulations, the application of restrains is crucial. Herein, we have used six relative orthogonal restrains with harmonic potentials that include one distance, two angles, and three dihedral angles restrain between the ligand and the protein with a force constant of 10 kcal mol$^{-1}$ Å$^{-2}$ [deg$^{-2}$]. At each $\lambda$, 10–30 ns of decoupling simulation via replica-exchange (*Affentranger et al., 2006*) were obtained to compute the $\Delta G_{bind}^{o}$ over a well-converged trajectory.

## Molecular dynamics setup

In this study, MD simulations were applied to compute the binding free-energy simulations via DDM. GROMACS-2018.8 (*Abraham et al., 2015*) was used as an MD engine for all simulations. The tleap module of AMBER16 was used to add the missing hydrogen atoms to the kinase enzymes. The system was solvated explicitly using TIP3P water boxes (*Jorgensen et al., 1983*) that extended at least 10 Å from the center of the system in each direction. The topology file for the kinase enzyme was created using the amber forcefield ff14SB (*Maier et al., 2015*). The AM1-BCC charge model (*Jakalian et al., 2002*) and general amber force field 2 (GAFF2) (*Wang et al., 2004*) were employed to parametrize different inhibitors used in this study. The overall charge of the system was maintained by adding a suitable number of counterions in each system. During the simulations, electrostatic interactions were computed using the particle mesh Ewald method (*Essmann et al., 1995*) with a cutoff and grid spacing of 10.0 and 1.0 Å, respectively. The NPT (constant Number of particles, Pressure, and Temperature) ensembles with a time step of 2 fs was used in the simulations.

## Benchmarking calculations for absolute binding free-energy simulations

Accurate prediction of the ABFE difference between the Apo and Holo state of a protein is extremely important to achieve from force field-based MD simulations (*Lin et al., 2013*; *Lovera et al., 2012*). In this study, we used type-II inhibitors as probes to estimate $\Delta G_{reorg}$ via ABFE simulations, which is the excess free-energy between experimentally determined binding affinity and $\Delta G_{bind}$ calculated from ABFE. Target kinases, i.e., MAPK14, CDK2, and JNK1 bound with type-I and I 1/2 inhibitors in the active conformation states (where $\Delta G_{reorg}$ is expected to be close to zero) have been regularly used by the computational community as benchmark systems for absolute or relative binding free energy calculations (*Wang et al., 2015*; *Lee et al., 2020*; *Goel et al., 2021*; *Kuhn et al., 2020*; *Gapsys et al., 2019*; *Khalak et al., 2021*). Benchmarking calculations over multiple protein-ligand complexes show close agreement between calculated ($\Delta G_{bind}$) and experimental ($\Delta G_{exp}$) terms (*Table 2*, *Figure 2—figure supplement 1*). In the later part of benchmarking studies, we have included ABL1 bound to type-II inhibitors in the DFG-out/folded activation loop state. NMR studies have shown experimentally that ABL1 has a $\Delta G_{reorg}$ of 1.2 kcal/mol (*Xie et al., 2020*), which is consistent with our range of estimates of $\Delta G_{reorg}$ for this kinase (*Table 2*).

## Potts-guided target selection for absolute binding free-energy simulations

Our Potts threaded-energy calculations were used alongside experimental type-II binding data from the large-scale assay by *Davis et al., 2011* to identify kinase targets that are likely to have very large or very small $\Delta G_{reorg}$. As described in the main text, all-atom MD simulations to calculate ABFEs of type-II inhibitors can be used alongside experimental binding affinities to calculate the free-energy

---

**Table 2.** Summary of benchmarking results.

All reported values are in units of kcal/mol. The root mean square error (RMSE) and mean unsigned error (MUE) were calculated with respect to the linear model $\Delta G_{bind}^{ABFE} + \Delta G_{reorg}$, where $\Delta G_{reorg}$ is calculated as $\left\langle \Delta G_{exp} - \Delta G_{bind}^{ABFE} \right\rangle$. The average difference between $\Delta G_{exp}$ and $\Delta G_{bind}^{ABFE}$ is shown in the last column.

| Kinase | # Compounds | MUE | RMSE | $\left| \left\langle \Delta G_{exp} - \Delta G_{bind}^{ABFE} \right\rangle \right|$ |
|---|---|---|---|---|
| CDK2 | 6 (type-I) | 0.71 | 1.03 | 1.89 |
| JNK1 | 5 (type-I) | 0.37 | 0.44 | 1.29 |
| MAPK14 | 9 (type-1$^{1}$/$_{2}$) | 0.47 | 0.64 | 0.2 |
| ABL1 | 6 (type-II) | 1.47 | 1.57 | 1.26 |

cost for a kinase to reorganize to the DFG-out/folded activation loop conformation. This relation, $\Delta G_{reorg} = \Delta G_{exp} - \Delta G_{bind}$(ABFE), gives the free-energy cost to reorganize in physical energy units (kcal/mol) and can be used to approximate a scale for the Potts statistical energy differences provided one sample a sufficient range of $\Delta G_{reorg}$ and $\Delta E_{Potts}$. However, ABFE simulations are much more computationally demanding than the Potts threading calculation, which we sought to mitigate by choosing kinase simulation targets which are likely to provide a strong signal, guided by the Potts model. We direct the reader to *Table 1*, which contains the Potts penalties and type-II hit rates for the targets of interest. For comparison, *Figure 1A* and *Figure 5* provide the overall distributions of hit rates and Potts penalties for TKs and STKs.

A significant challenge for our target selection was the limited availability of type-II inhibitors co-crystallized against STKs which have (a) Potts penalties and type-II hit rates that predict very high $\Delta G_{reorg}$, (b) experimental binding affinities available in the literature in the form of IC50, Ki, or Kd, (c) availability of protein-ligand co-crystallized structure(s), and (d) type-II inhibitor complex systems where the activation loop appears to have undergone a large-scale 'folding' conformational change relative to the active 'extended' conformation. STK complexes that satisfy all four criteria appear to be sparse, which is consistent with the notion that kinases with large reorganization penalties are more difficult to crystallize in the classical DFG-out conformation (*Haldane et al., 2016*). However, for some STKs with very high Potts threaded-energy penalties (e.g., MELK) there has been significant medicinal chemistry efforts to design potent type-II inhibitors and structurally characterize their complexes using x-ray crystallography. Using type-II co-crystal structures that cover five different STKs with high Potts penalties (*Table 1*) and five different TKs with low Potts penalties, we were able to sample a wide range of $\Delta G_{reorg}$ from a total of 45 ABFE simulations covering 45 type-II inhibitor complexes and 10 different kinase targets. These simulations and subsequent calculations of $\Delta G_{reorg}$ for each kinase resulted in a strong correlation with the Potts threaded energy scores (*Figure 5*), allowing us to establish a scale for the Potts energies in kcal/mol. We have provided detailed results from the ABFE simulations of these 10 kinase targets in the form of supplementary figures (*Figure 3—figure supplement 1*) where the average $\Delta G_{reorg}$ for each kinase is visualized as the y-intercept of a linear regression with the slope constrained to one.

## Data availability

Values of $\Delta G_{bind}^{ABFE}$ from all ABFE simulations described in this work, including benchmarking calculations (94 simulations in total), are provided in the form of supplementary tables (*Figure 2—source data 1* and *Figure 3—source data 1*). Potts $\Delta Es$, type-II hit rates computed from *Shan et al., 2013*, the identity of gatekeeper residues and corresponding van der Waals volumes in Å (*Stancik et al., 2018*), and the classification of human kinases as TKs or STKs were provided in a separate supplementary table (*Figure 1—source data 1*).

## Acknowledgements

This research was supported by National Institutes of Health grant number R35-GM132090, and by NIH Computer Equipment Grant (OD020095). Gratitude is also expressed to the OWLSNEST high performance cluster at Temple University for its computing support in this project. We thank Shima Arasteh for helpful discussions related to kinase conformational states and alchemical free-energy simulations.

## Additional information

### Funding

| Funder | Grant reference number | Author |
|---|---|---|
| National Institutes of Health | R35-GM132090 | Joan Gizzio<br>Abhishek Thakur<br>Allan Haldane<br>Ronald M Levy |

| Funder | Grant reference number | Author |
|--------|------------------------|--------|
| National Institutes of Health | OD020095 | Joan Gizzio<br>Abhishek Thakur<br>Allan Haldane<br>Ronald M Levy |

The funders had no role in study design, data collection and interpretation, or the decision to submit the work for publication.

## Author contributions

Joan Gizzio, Abhishek Thakur, Conceptualization, Data curation, Software, Formal analysis, Validation, Investigation, Visualization, Writing - original draft, Writing - review and editing; Allan Haldane, Conceptualization, Resources, Data curation, Software, Formal analysis, Supervision, Investigation, Methodology, Writing - review and editing; Ronald M Levy, Conceptualization, Resources, Supervision, Funding acquisition, Investigation, Methodology, Project administration, Writing - review and editing

## Author ORCIDs

Joan Gizzio http://orcid.org/0000-0002-4127-2294
Abhishek Thakur http://orcid.org/0000-0002-4827-7602
Allan Haldane http://orcid.org/0000-0002-8343-1994
Ronald M Levy http://orcid.org/0000-0001-8696-5177

## Decision letter and Author response

Decision letter https://doi.org/10.7554/eLife.83368.sa1
Author response https://doi.org/10.7554/eLife.83368.sa2

## Additional files

### Supplementary files

- MDAR checklist

### Data availability

Our computational study makes use of experimental data from the literature, which we extracted and curated manually rather than relying on any specific database. Any experimental data used can be found in our supporting information in the form of tables alongside appropriate citations. A large set of experimental "hit rates" were derived from binding affinities available from *Davis et al., 2011*. The data used to generate various plots in the main text can be found in tables throughout the supporting information, as well as a distinct "supplementary table" which we provide. The Mi3-GPU (*Haldane and Levy, 2021*) source code required to reproduce the Potts model employed in this manuscript can be found at the following link: https://github.com/ahaldane/Mi3-GPU (*Haldane and avikbiswas, 2021*, v1.1, copy archived at swh:1:rev:b8fd4aa67bb2531fdc60e3a00fed6f80c8aceb49).

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
