## [Editor Report]

This important paper provides a convincing mechanism for the relative binding specificity of Type II inhibitors to kinases. The combination of a sequence-derived Potts model with experimental dissociation constants and calculated free energies of binding to the DFG-out state is highly compelling and goes beyond the current state-of-the-art. Given the importance of kinases in pathophysiological processes, the results will be of interest to a broad audience and, in addition, the combination of computational methods can be applicable to a wide variety of other biophysical processes that involve conformational rearrangements.

---

## [Decision Letter]

**Decision letter after peer review:**

Thank you for submitting your article "Evolutionary Divergence in the Conformational Landscapes of Tyrosine vs Serine/Threonine Kinases" for consideration by *eLife*. Your article has been reviewed by 3 peer reviewers, and the evaluation has been overseen by a Reviewing Editor and Amy Andreotti as the Senior Editor. The following individual involved in review of your submission has agreed to reveal their identity: Robert Best (Reviewer #1).

Essential revisions:

A description of the limitations of the work should be more prominent. You indeed should explicitly state your assumptions about the energy landscape containing two main states, DFG-in and DFG-out. In that context, the conformation of the C-helix should be discussed, even if it has no bearings on the final results.

*Reviewer #1 (Recommendations for the authors):*

– I was interested in whether the role of the gatekeeper residue would be also captured by the Potts models and what its relative contribution to overall free energy difference would be. For example if one computed an average δ δ E for mutating large to small gatekeepers, does this follow the expected trend of small gatekeepers favoring DFG-out, and what is the magnitude of the effect? And whether the effect depends on the context (STK vs TK).

– Showing the location of the 'gatekeeper' residue in the structure in Figure 1 or elsewhere would help the reader to visualize the location of this frequently discussed residue. Also, there is not much discussion of the gatekeeper after Figure 1, even though it is clear that it plays a role in inhibitor binding.

– Are there data for Type I inhibitors for TK's to show in Figure 3? If dG is ~0 between DFG-in and DFG-out, those would also lie close to the y=x line.

*Reviewer #2 (Recommendations for the authors):*

This paper focuses on an important topic. It explores how the activation loop conformations affect the type II inhibitor binding to Tyr and Ser/Thr kinases. The computational results agree with the available experimental data. It is a remarkably comprehensive, high quality paper. Some minor comments are as follows:

1. The work focuses on the classical "folded" DFG-out conformation of kinases. From the available PDB structures, most of DFG-out kinases have no well-defined "folded" activation loop structures. In many of them, the activation loops after "DFG" motif have no complete density, which indicates that they are flexible and can be disordered. Given that "folded" for protein structures sometimes means that they form α-helix or β strand, the authors may consider changing "folded" into other descriptors or removing it.

2. Smaller gatekeepers and the sequence of the activation loop contribute to the binding of type II inhibitors in the pocket. The gatekeepers have direct contacts with the inhibitors, while the activation loop does not. Can the authors comment on which one would be more important for determining the type II inhibitor binding?

3. The transition energies were calculated from the active DFG-in conformation to the inactive DFG-out conformation. Please note that the DFG-in conformation has two states as well, i.e., the inactive state (with aC-out) and the active state (with aC-in) (PMID: 35003591, PMID: 33971270). Some kinases barely show the DFG-out conformation (PMID: 32918948). The binding of the type II inhibitors in the pockets does not necessarily always go through the active DFG-in conformation (ac-in). It can also go through the inactive DFG-in conformation (aC-out) to DFG-out. Does the calculated energy data consider the alternative structural transitions?

4. In Figure 5, please provide more details about the collected 10,345 tyrosine kinases and 210,862 serine/threonine kinases.

5. How the activation loop sequences are aligned, and are they structurally validated?

*Reviewer #3 (Recommendations for the authors):*

I find the paper to be well-written and don't have any major critiques. However, my minor concern is about the presentation of the subject in the Introduction. For a reader who is not familiar with the topic, it would sound that the "DFG-out" is the major inactive conformation. And although the authors do talk about "Src-like" confirmation and present distribution of inactive structures, it is in the Methods section or in Supplementary material. The method has its limits and it is better to describe them directly in the Introduction. A lot of information that would be helpful for a reader to understand the scope and the importance of the paper is moved to the Methods section which is almost 14 pages long, compared to 4 pages of Introduction. I would advise the authors to reconsider their Methods and move some important concepts into the Introduction.

---

## [Author Response]

Essential revisions:A description of the limitations of the work should be more prominent. You indeed should explicitly state your assumptions about the energy landscape containing two main states, DFG-in and DFG-out. In that context, the conformation of the C-helix should be discussed, even if it has no bearings on the final results.

We thank the editors and reviewers for their useful feedback. As discussed below in our responses to the reviewers, we have moved the discussion of the αC-helix and its role in the conformational landscape (αC-in ↔ αC-out) to the introduction. We have also made it clear in the introduction that our Potts threading calculations are done with a two-state model (active DFG-in αC-in and inactive classical DFG-out). Even though the αC-helix plays a role in the transition pathway between active DFG-in and inactive DFG-out, its conformation is largely the same in the initial and final states. Therefore, the different conformations that can be adopted by the αC-helix do not significantly affect the Potts threaded energies between the initial and final states, which were calculated for the change in threaded energy between the active DFG-in and inactive DFG-out basins only.

Reviewer #1 (Recommendations for the authors):– I was interested in whether the role of the gatekeeper residue would be also captured by the Potts models and what its relative contribution to overall free energy difference would be. For example if one computed an average δ δ E for mutating large to small gatekeepers, does this follow the expected trend of small gatekeepers favoring DFG-out, and what is the magnitude of the effect? And whether the effect depends on the context (STK vs TK).

The reviewer brings up an excellent point regarding the effect of the gatekeeper on the conformational landscape, and it is interesting to ask whether our Potts calculations would capture this effect in the context of type-II binding. When comparing the two conformational ensembles or free energy basins studied in our paper (active DFG-in aC-in and “classical” DFG-out) we find that the gatekeeper’s environment, measured by residue-residue contacts, is not significantly different in the two ensembles or basins. Therefore, our calculation of ΔE from the Potts model suggests that gatekeeper size does not directly contribute to the free energy difference between the two basins. Despite this, in Figure 1a we show that the gatekeeper size likely does effect type-II binding. To rationalize this, we note that the Potts calculations focus only on the conformational reorganization component of type-II binding. Meanwhile, the free-energy of binding to the already-reorganized receptor will include the effects of gatekeeper size due to direct interactions between the type-II inhibitors and the gatekeeper residue. As the reviewer mentions there is a “... trend of small gatekeepers favoring classical DFG-out” in the sense that most kinase structures crystallized in classical DFG-out in the PDB have small gatekeepers. But the very large majority of those structures are co-crystallized with a type-II inhibitor. Our results suggest this trend is related to the more favorable interactions of type-II inhibitors with smaller gatekeeper residues in the reorganized DFG-out binding pocket, rather than the effect of gatekeeper size on the reorganization free energy in the absence of bound type-II inhibitors.

– Showing the location of the 'gatekeeper' residue in the structure in Figure 1 or elsewhere would help the reader to visualize the location of this frequently discussed residue. Also, there is not much discussion of the gatekeeper after Figure 1, even though it is clear that it plays a role in inhibitor binding.

We thank the reviewer for the suggestion to modify Figure 1. We have made changes in the Figure by showing the position of the gatekeeper residue to make it easier for readers to visualize. We would also like to address the reviewer’s concern regarding not much discussion of the gatekeeper residue in the manuscript after Figure 1. Our results using the Potts model to estimate the reorganization free energy from the active DFG-in aC-in basin to the classical DFG-out inactive basin are most strongly influenced by the sequence variation of the activation loop and the motifs that it directly interacts with, which does not include the gatekeeper residue. However, as we pointed out in Figure 1a, the type-II hit rates of kinases (especially TKs) with large vs small gatekeepers are significantly different, consistent with current understanding in the literature regarding the role of the gatekeeper in type-II binding. Dissecting the effect of the direct interactions between the gatekeeper residue and type-II inhibitors on binding affinities is outside the scope of our manuscript but will be an interesting subject of future research.

– Are there data for Type I inhibitors for TK's to show in Figure 3? If dG is ~0 between DFG-in and DFG-out, those would also lie close to the y=x line.

It is true that Type-I inhibitors vs TKs would also lie close to the y=x line. Our reason for plotting type-I inhibitors on the same axis as type-II inhibitors for STKs is because of the large difference between ΔGexp and ΔGbind for type-II inhibitors, which we hypothesize is related to the large ΔGreorg between DFG-in and DFG-out for STKs. This required us to rule out other sources of unexpected systematic error unrelated to the DFG-in ↔ DFG-out equilibrium for STKs, hence this is why we have shown the plot of ΔGreorg ~ 0 for type-I inhibitors vs STKs but not TKs.

Reviewer #2 (Recommendations for the authors):This paper focuses on an important topic. It explores how the activation loop conformations affect the type II inhibitor binding to Tyr and Ser/Thr kinases. The computational results agree with the available experimental data. It is a remarkably comprehensive, high quality paper. Some minor comments are as follows:1. The work focuses on the classical "folded" DFG-out conformation of kinases. From the available PDB structures, most of DFG-out kinases have no well-defined "folded" activation loop structures. In many of them, the activation loops after "DFG" motif have no complete density, which indicates that they are flexible and can be disordered. Given that "folded" for protein structures sometimes means that they form α-helix or β strand, the authors may consider changing "folded" into other descriptors or removing it.

We thank the reviewer for pointing out this potential confusion. We have added clarification for this nomenclature in the Introduction, stating that “folded activation loop” in our nomenclature collectively refers to activation loops which have undergone a large conformational change of ~17A away from the active/extended state. This nomenclature has been used previously in the literature to describe the activation loop conformation in the classical DFG-out state.

2. Smaller gatekeepers and the sequence of the activation loop contribute to the binding of type II inhibitors in the pocket. The gatekeepers have direct contacts with the inhibitors, while the activation loop does not. Can the authors comment on which one would be more important for determining the type II inhibitor binding?

Indeed, gatekeeper size has a strong effect on type-II binding via direct interactions with the inhibitor; for example, it has been shown by other authors that mutating from small to large gatekeepers (e.g. T315I in Abl) increases the binding free-energy to the DFG-out kinase by >3 kcal/mol. Our observation in Figure 1a that gatekeeper size is correlated with type-II inhibitor hit appears consistent with this. However, we also show in Figure 1a that differences in gatekeeper size do not explain the large shift in type-II hit rates between TKs and STKs. From the results of our Potts model calculations, we proposed that residue substitutions in the activation loop rather than gatekeeper are largely responsible for this shift which is associated with the free energy cost to reorganize from the active DFG-in aC-in conformational basin to the inactive classical DFG-out basin.

3. The transition energies were calculated from the active DFG-in conformation to the inactive DFG-out conformation. Please note that the DFG-in conformation has two states as well, i.e., the inactive state (with aC-out) and the active state (with aC-in) (PMID: 35003591, PMID: 33971270). Some kinases barely show the DFG-out conformation (PMID: 32918948). The binding of the type II inhibitors in the pockets does not necessarily always go through the active DFG-in conformation (ac-in). It can also go through the inactive DFG-in conformation (aC-out) to DFG-out. Does the calculated energy data consider the alternative structural transitions?

The calculated Potts threaded energies consider two conformational basins only (active DFG-in aC-in, and inactive classical DFG-out). We have clarified this in the introduction and point out that the good correlation between ΔGreorg and Potts energies calculated in this way suggests the two-state model is sufficient for distinguishing the contribution of the reorganization free energy from the active DFG-in aC -in basin to the inactive classical DFG-out basin between TKs and STKs. As suggested in the literature, the inactive DFG-in aC-out conformation(also called Src-like inactive) is likely an intermediate on the pathway from DFG-in active to classical DFG-out inactive. This intermediate state may well affect the kinetics of the DFG-in active to classical DFG-out inactive transition, but not the thermodynamics of the transition between the initial DFG-in active and final classical DFG-out inactive free energy basins.

4. In Figure 5, please provide more details about the collected 10,345 tyrosine kinases and 210,862 serine/threonine kinases.

We thank the reviewer for pointing this out. In the revised version of the manuscript, we have added details about the sequence diversity and phylogenetics of these kinases in the Figure 5 caption.

5. How the activation loop sequences are aligned, and are they structurally validated?

This is an excellent question given that sequence alignments of flexible loop regions can be difficult to construct. The kinase domain sequences including activation loops were aligned starting from a PFAM seed, which was used with HHblits to align evolutionarily related kinases from the UniProt database. The aligned activation loops are 20 residues long with gaps and insertions assigned accordingly – typically this occurs near the center of the activation loop where there is significant flexibility and low homology. In general the activation loop alignments are most accurate near the N-terminal and C-terminal residues which are more evolutionarily conserved and not as flexible as the center-region of the activation loop. Our results with the Potts model are derived primarily from accurately aligned regions of the activation loop, which we have confirmed via structural superposition of important TKs and STKs.

Reviewer #3 (Recommendations for the authors):I find the paper to be well-written and don't have any major critiques. However, my minor concern is about the presentation of the subject in the Introduction. For a reader who is not familiar with the topic, it would sound that the "DFG-out" is the major inactive conformation. And although the authors do talk about "Src-like" confirmation and present distribution of inactive structures, it is in the Methods section or in Supplementary material. The method has its limits and it is better to describe them directly in the Introduction. A lot of information that would be helpful for a reader to understand the scope and the importance of the paper is moved to the Methods section which is almost 14 pages long, compared to 4 pages of Introduction. I would advise the authors to reconsider their Methods and move some important concepts into the Introduction.

As the reviewer suggests, we have moved the introduction of the Src-like inactive conformation and a description of other features of the kinase conformational landscape from the Methods section to the introduction. We also note that while the major inactive conformation in the absence of inhibitors may differ from kinase to kinase, the only inactive conformation compatible with the type-II binding mode is classical DFG-out, which is why we have placed significant focus on this free-energy basin. We hope that our revised introduction will give a more holistic view of the conformational landscape in the absence of inhibitors, but we emphasize that the classical DFG-out conformational free energy basin is the one which is populated when type-II inhibitors are bound.